# Electron Bernstein waves driven by electron crescents near the electron diffusion region

W.Y. Li [1,2,3]*, D.B. Graham[2]*, Yu.V. Khotyaintsev [2], A. Vaivads [4], M. André[2], K. Min [5], K. Liu[6], B.B. Tang [1], C. Wang[1], K. Fujimoto[7], C. Norgren[8], S. Toledo-Redondo[9,10], P.-A. Lindqvist [4], R.E. Ergun[11], R.B. Torbert [12], A.C. Rager[13,14], J.C. Dorelli[14], D.J. Gershman[14,15], B.L. Giles[14], B. Lavraud[9], F. Plaschke [16], W. Magnes[16], O. Le Contel [17], C.T. Russell[18] & J.L. Burch [19]

The Magnetospheric Multiscale (MMS) spacecraft encounter an electron diffusion region (EDR) of asymmetric magnetic reconnection at Earth's magnetopause. The EDR is characterized by agyrotropic electron velocity distributions on both sides of the neutral line. Various types of plasma waves are produced by the magnetic reconnection in and near the EDR. Here we report large-amplitude electron Bernstein waves (EBWs) at the electron-scale boundary of the Hall current reversal. The finite gyroradius effect of the outflow electrons generates the crescent-shaped agyrotropic electron distributions, which drive the EBWs. The EBWs propagate toward the central EDR. The amplitude of the EBWs is sufficiently large to thermalize and diffuse electrons around the EDR. The EBWs contribute to the cross-field diffusion of the electron-scale boundary of the Hall current reversal near the EDR.

---

[1] State Key Laboratory of Space Weather, National Space Science Center, Chinese Academy of Sciences, 100190 Beijing, China. [2] Swedish Institute of Space Physics, SE-75121 Uppsala, Sweden. [3] State Key Laboratory of Lunar and Planetary Sciences, Macau University of Science and Technology, Macau, China. [4] Division of Space and Plasma Physics, School of Electrical Engineering and Computer Science, KTH Royal Institute of Technology, SE-11428 Stockholm, Sweden. [5] Department of Astronomy and Space Science, Chungnam National University, Daejeon 34134, Republic of Korea. [6] Department of Earth and Space Sciences, Southern University of Science and Technology, 518055 Shenzhen, China. [7] School of Space and Environment, Beihang University, 100191 Beijing, China. [8] Department of Physics and Technology, University of Bergen, 5020 Bergen, Norway. [9] Institut de Recherche en Astrophysique et Planétologie, Université de Toulouse, CNRS, UPS, CNES, 31028 Toulouse, France. [10] Department of Electromagnetism and Electronics, University of Murcia, 30003 Murcia, Spain. [11] Laboratory of Atmospheric and Space Physics, University of Colorado, Boulder, CO 80303, USA. [12] Space Science Center, University of New Hampshire, Durham, NH 03824, USA. [13] Catholic University of America, Washington, DC 20064, USA. [14] NASA Goddard Space Flight Center, Greenbelt, MD 20771, USA. [15] Department of Astronomy, University of Maryland, College Park, MD 20742, USA. [16] Space Research Institute, Austrian Academy of Sciences, 8042 Graz, Austria. [17] Laboratoire de Physique des Plasmas, CNRS/Ecole Polytechnique/Sorbonne Université/Univ. Paris Sud/ Observatoire de Paris, 75252 Paris, France. [18] Department of Earth and Space Sciences, University of California, Los Angeles, CA 90095, USA. [19] Southwest Research Institute, San Antonio, TX 78238, USA. *email: wyli@spaceweather.ac.cn; dgraham@irfu.se

**M**agnetic reconnection is a fundamental and universal process, which transfers energy stored in the magnetic field to kinetic energy of charged particles[1,2]. Magnetic reconnection powers eruptive processes in space and laboratory plasmas. Earth's magnetosphere provides a unique environment to study magnetic reconnection by analyzing in situ spacecraft measurements. NASA's MMS mission was designed to resolve the particles and fields at electron scales. The goal of MMS is to investigate the EDR, which is the core region of reconnection where the magnetic field lines break and reconnect[1,3]. Reconnection at the dayside magnetopause is asymmetric due to the large plasma and magnetic field differences between the magnetosheath and the magnetosphere[4,5]. Numerical simulations and MMS observations of asymmetric reconnection show that the crescent-shaped agyrotropic electrons can be found on both magnetospheric and magnetosheath sides of the neutral line[3,6–9].

Waves are suggested to generate anomalous resistivity and plasma diffusion, potentially enabling magnetic fields to break and reconnect[10]. Various types of waves produced by reconnection have been reported outside of EDRs[10–16]. In and near EDRs, MMS has observed upper-hybrid (UH)[9,17,18] and whistler[19] waves and low-frequency turbulent fluctuations[20]. Among these waves, the high-frequency electrostatic UH waves are driven by the agyrotropic electron beams near an EDR encounter. The amplitudes of the upper-hybrid waves are sufficiently large to interact with electrons and contribute to electron diffusion and scattering near the EDR[9]. Here we report an MMS observation of large-amplitude EBWs driven by electron crescents near an EDR. The EBWs have sufficient large amplitude to thermalize and diffuse electrons around the EDR. The EBWs contribute to the cross-field diffusion of the electron-scale boundary.

## Results

**Electron diffusion region encounter by MMS.** The EDR event was encountered near Earth's subsolar magnetopause on December 24, 2016. MMS were located at [9.4, 3.7, 1.6] $R_E$ (Earth radii) in geocentric solar ecliptic (GSE) coordinates. We use

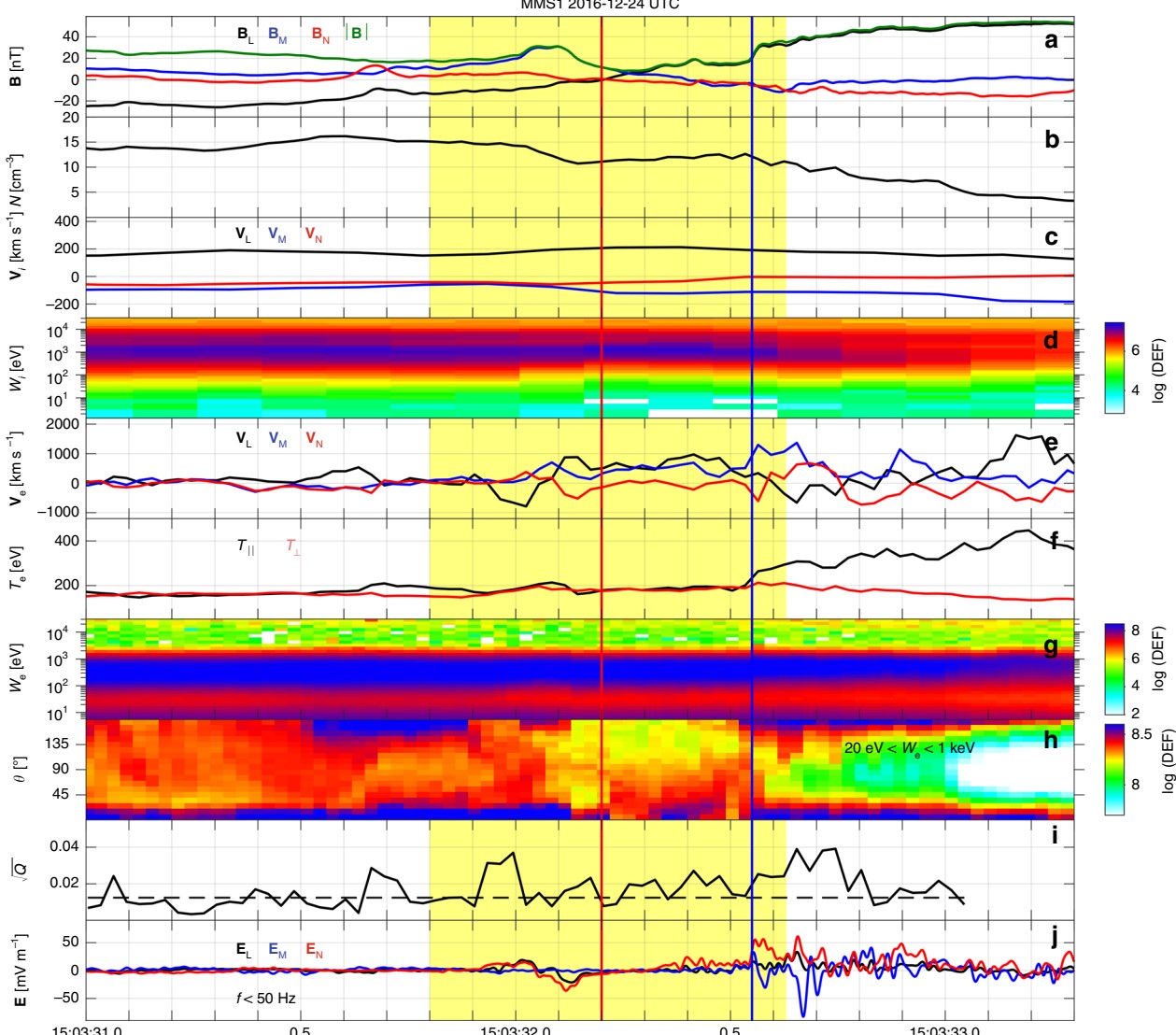

**Fig. 1 Magnetopause crossing observed by MMS1. a** B. **b** Number density $N$. **c** $V_i$. **d** Ion differential energy flux (color scale, in unit of keV s$^{-1}$ cm$^{-2}$ sr$^{-1}$ keV$^{-1}$). **e** $V_e$. **f** Electron $T_{||}$ and $T_{\perp}$. **g** Electron differential energy flux. **h** Electron pitch-angle distribution between 20 eV and 1 keV. **i** Agyrotropy measure $\sqrt{Q}$[27] with a background of 0.012 (black dashed line). **j** E with frequencies $f < 50$ Hz. The vectors are all presented in LMN coordinate system. The red and blue vertical lines represent the neutral line and the magnetospheric separatrix, respectively. The agyrotropy measure $\sqrt{Q}$ with $N < 5$ cm$^{-3}$ is neglected. A detailed overview of the yellow-shaded region is presented in Fig. 3.

magnetic field **B** data from the fluxgate magnetometer[21] and the search-coil magnetometer[22]. The electric field **E** data are from the electric field double probes[23,24]. The particle data are from the fast plasma investigation (FPI)[25]. All data presented are in high-resolution burst mode. The burst-mode electron data are sampled at 30 ms resolution. The vectors are shown in boundary-normal (LMN) coordinates based on minimum variance analysis (MVA)[26] of **B** over the magnetopause crossing, unless otherwise stated. Here, **L** = [0.09, 0.08, 0.99] is along the reconnecting magnetic field direction, **M** = [0.23, −0.97, 0.06] is the out-of-plane direction, and **N** = [0.97, 0.22, −0.10] is the normal direction in GSE coordinates.

Figure 1 presents an overview of the magnetopause crossing observed by MMS1, which is sketched in Fig. 2. The crossing from the reconnection outflow to the magnetosphere is characterized by a $B_L$ reversal (see Fig. 1a), a decrease in plasma density (see Fig. 1b), and an ion outflow ($V_{iL} > 0$) (see Fig. 1c). The magnetic shear angle between the magnetosheath and magnetosphere is about 162°, and so the guide field (~12 nT) in this event is relatively weak. In Fig. 1, the neutral line is located at the time indicated by the red vertical line, determined by the $B_L$ zero-crossing point. On the magnetosheath side of the neutral line, MMS1 observes a strong increase in $B_M$, which corresponds to the Hall magnetic field[5,28,29]. The magnetospheric separatrix is located at the time indicated by the blue vertical line, and is characterized by a fast electron flow along **M** direction (see Fig. 1e), an increase in $T_{e,\parallel}$ (see Fig. 1f), a rapid increase of $B_L$ (see Fig. 1a), and an $E_N$ (see Fig. 1j). After the separatrix crossing,

the enhanced electron fluxes along the directions parallel and antiparallel to **B** (see Fig. 1h) are trapped electrons in the magnetospheric inflow region[30,31]. The parameter $\sqrt{Q}$ provides a measure of agyrotropy based on the magnitude of the off-diagonal terms of the electron pressure tensor in field-aligned coordinates[27]. Typical values are about 0.1 around the electron diffusion region at the magnetopause[8,9,14]. As shown in Fig. 1i, electron velocity distributions with enhanced agyrotropies are observed over the magnetopause crossing, including the magnetosheath and magnetospheric sides of the neutral line. All these features indicate that MMS are in or near an electron diffusion region of the magnetopause reconnection[3,5–7,9,20,31–36].

Figure 3a–d presents a detailed overview of the yellow-shaded region in Fig. 1. Figure 3b shows the electric current **J** perpendicular to **B** calculated from the particle moments[37,38]. The strong Hall magnetic field $B_M$ (see Fig. 3a) is located at the $J_\perp$ reversal from **L** to −**L**, which is mainly carried by an electron flow reversal from −**L** to **L** (see Fig. 1e). Here, the electron thermal gyroradius is $\rho_e \sim 1.7$ km, similar to the electron inertial length (~1.4 km). The normal speed $V_N \sim 50$ km s$^{-1}$ of the magnetopause motion is estimated from multispacecraft timing analysis[39] of $B_L$ around the neutral line. The peak-to-peak time duration of the Hall-current region is about 0.12 s, which corresponds to a normal scale of 3.5 $\rho_e$.

We observe electron distributions with enhanced agyrotropies on both sides of the Hall current reversal (see Fig. 1i). Figure 3e, h shows the electron distributions at times indicated by the two yellow bars in Fig. 3a, c, respectively, which are the Hall current peaks. MMS1 observes dense crescents in the plane perpendicular to **B** and close to the **E** × **B** directions (more details in the section "Electron velocity distribution functions"). The electron crescents point close to the −**L** and **L** directions (shown by black lines in Fig. 3e, g) on the magnetosheath and the magnetospheric edges of the Hall **B**, respectively. The electron beams parallel to **B** (see Fig. 3f, h), seen as the enhanced energy fluxes parallel to **B** in Fig. 3d, are the magnetosheath inflow electrons moving towards the central diffusion region. Here, **B** is dominated by the Hall magnetic field, $B_M$.

**Large-amplitude EBWs.** Large-amplitude high-frequency waves are observed at the electron-scale boundary of the Hall current reversal. Figure 4b shows the waveform of the high-frequency $E_\perp$ and $E_\parallel$ (with respect to **B**) around the point where $V_\perp = E \times B/B^2$ changes sign (curves in Fig. 4c). In the yellow-shaded region, we can see that $E_\perp \gg E_\parallel$. $E_{\perp,L}$ is much larger than the other perpendicular component $E_{\perp 2}$, making the $E_{\perp,L}$ direction about 19° away from the **L** direction. We see distinct spectral peaks separated in frequency by approximately the electron cyclotron frequency $f_{ce}$ (see Fig. 4d, e). The power spectrum of $E_\perp$ within the yellow-shaded region is presented in Fig. 4f. The largest wave powers are observed in a range between 4.5 and 8.5 kHz, i.e., between the fifth and the ninth harmonics of $f_{ce}$ (~860 Hz).

Figure 4g, h presents the hodograms of $E_{max}$ versus $E_{int}$ and $E_{max}$ versus $E_{min}$, respectively. Here, $E_{max}$, $E_{int}$, and $E_{min}$ are the electric fields in the maximum ([−0.97, −0.16, 0.19], in LMN), intermediate ([0.21, −0.11, 0.98]), and minimum ([−0.13, 0.98, 0.14]) variance directions based on MVA of the **E** waveform high-pass filtered above 50 Hz, so all electron Bernstein modes are included. The high-frequency waves have a well-defined maximum variance direction, which is 166° away from the reconnecting magnetic field direction **L** and approximately 84° away from **B**. In addition, the wave fluctuations exhibit approximately linear polarization. We also observe extremely weak (~0.02 nT) **B** fluctuations (see Fig. 4e) due to a small angle between the wave propagation direction and **E** fluctuation

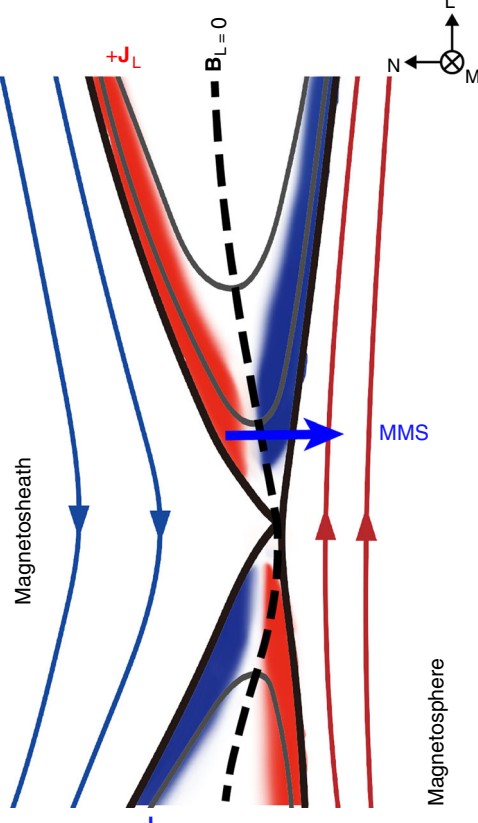

**Fig. 2 Sketch of asymmetric reconnection with $J_L$.** The black dashed curve shows the neutral line (defined where $B_L = 0$), and the blue arrowed line denotes the MMS trajectory near the electron diffusion region. The large-amplitude electron Bernstein waves are observed on the magnetosheath side of the neutral line.

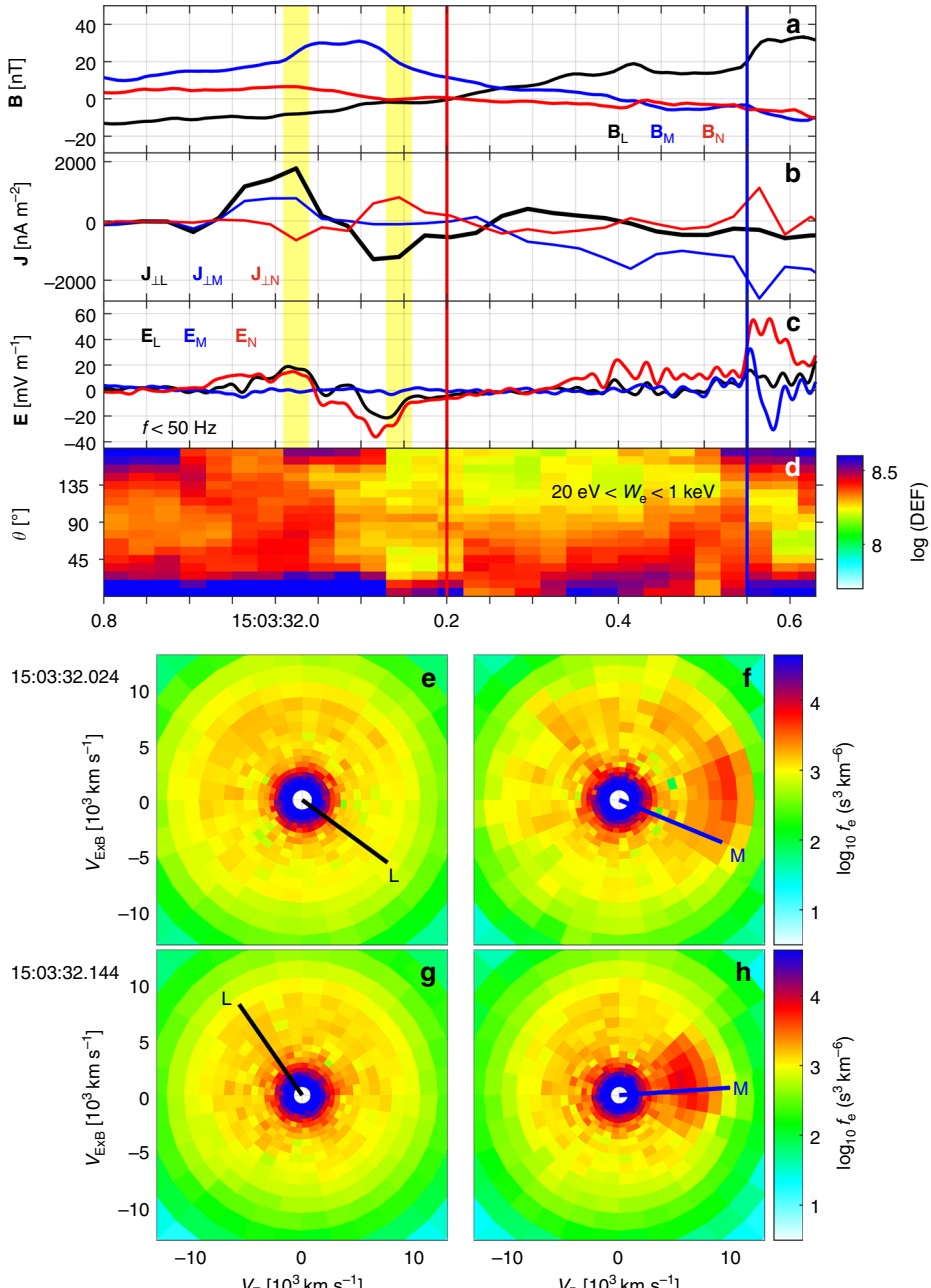

**Fig. 3 Electron-scale boundary of the Hall current reversal. a B**. **b** Perpendicular components of **J** calculated by high-resolution FPI particle moments.
**c E** with $f < 50$ Hz. **d** Electron pitch-angle distribution between 20 eV and 1 keV. The red and blue vertical lines represent the neutral line and the
magnetospheric separatrix, respectively. **e–h** Electron distributions at times indicated by the two vertical yellow bars in (**a–c**), respectively. **J** calculated by
FPI data agrees well with the current estimated by the curlometer method[40], which has been verified by the current event and other MMS events[37,38]. The
vectors are all presented in LMN coordinate system, and the **L** and **M** directions normalized by $10^4$ km s$^{-1}$ are projected on (**e–h**).

direction. All these observed properties of the high-frequency
quasi-electrostatic waves are consistent with the characteristics of
electron Bernstein waves[41] expected in an over-dense plasma.
Therefore, $\mathbf{E}_{max}$ direction provides a good estimation of the wave
vector direction $\hat{\mathbf{k}}$. In space plasma environments, the EBWs can
also be observed around the bow shock[42] and inside the
magnetosphere[43,44]. In fusion devices, the EBWs are widely used
to heat plasmas[45].

The EBWs are observed for about 17 ms, corresponding to a
spatial scale of 0.5 $\rho_e$ normal to the magnetopause. We adopt the
7.5 ms electron distribution function data[46] for detailed analysis
of the generation of EBWs. In this case, the 7.5 ms electron data

do not change substantially around the EBWs interval, which
means that the 7.5 ms data provide nonaliased electron distribu-
tion functions before and during the large-amplitude EBWs. The
electron distributions at times indicated by the three green lines
in Fig. 5b are presented in Fig. 6a–c. Figure 6a shows the electron
distribution just before the large-amplitude EBWs. The crescent-
shaped electron distributions are oriented close to the $\mathbf{E} \times \mathbf{B}$
direction (see Fig. 7). The crescents have a clear positive gradient
in the direction with the largest phase-space density (black curve
in Fig. 6d), which is 7.3° away from the $\mathbf{E}_{max}$ direction (blue line
in Fig. 6a) of the EBWs. The typical speed of the electrons
constituting the crescents is $10^4$ km s$^{-1}$, and the number density

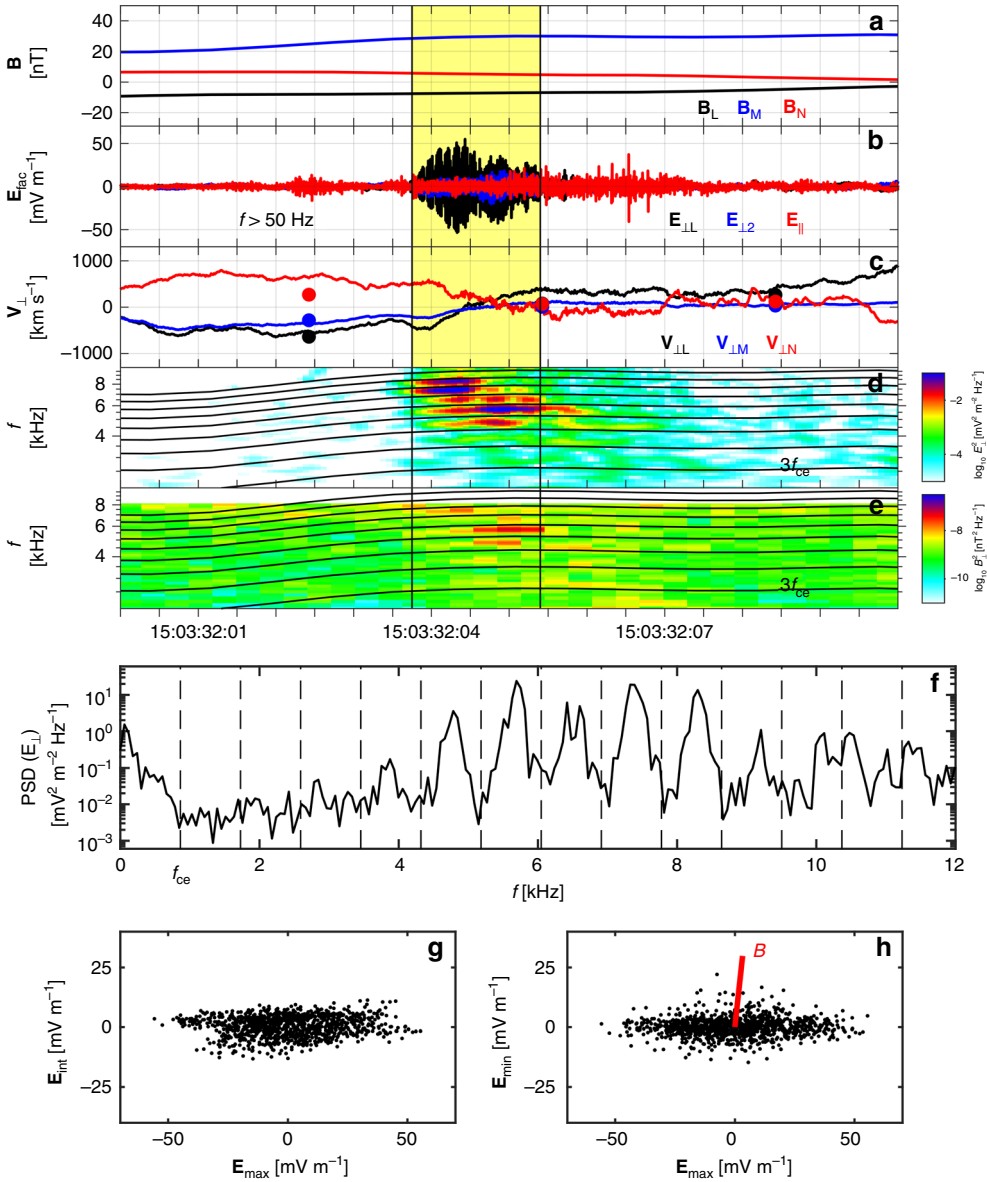

**Fig. 4 EBWs observed near EDR. a** B. **b** Perpendicular and parallel components of the high-frequency **E** with $f > 50$ Hz. (**c**) $\mathbf{E} \times \mathbf{B}/B^2$ (lines) and 30-ms resolution $\mathbf{V}_{e,\perp}$ (dots). **d** Power spectrogram of $\mathbf{E}_\perp$. **e** Power spectrogram of **B**. The electron cyclotron harmonic frequencies are plotted in (**d**, **e**). The yellow-shaded region from 15:03:32.037 to 15:03:32.054 UTC highlights the **E** fluctuations with amplitudes larger than $E_{peak}/e^2$, where $E_{peak}$ is the peak of the fluctuating **E**, and $e \sim 2.718$ is the Euler identity. **f** Power spectrum of $\mathbf{E}_\perp$ within the yellow-shaded region. The average $f_{ce}$ and the harmonic frequencies are over-plotted. **g, h** Hodograms of $\mathbf{E}_{max}$ versus $\mathbf{E}_{int}$, and $\mathbf{E}_{max}$ versus $\mathbf{E}_{min}$. The red line in (**h**) denotes the **B** direction. During the EBWs observations, the electron plasma frequency $f_{pe} \sim 32.9$ kHz is well above the electron cyclotron frequency $f_{ce} \sim 860$ Hz.

of the crescents is ∼54% of the total number density. The crescent energy density is about four orders of magnitudes larger than the maximum wave energy density. It is likely that the crescent-shaped electrons are the driving source of the large-amplitude EBWs. The electron speed with the positive gradient, $8 \times 10^3$ km s$^{-1}$, provides a good estimation of the phase velocity $\mathbf{V}_{ph}$ of the EBWs. The frequency of peak wave power (5.7 kHz, between $6 f_{ce}$ and $7 f_{ce}$, see Fig. 4f) corresponds to a wavelength of 1.4 km, which is comparable to $\rho_e$. The direction with the largest crescent phase-space density (black line in Fig. 6a), [−0.97, −0.14, 0.15] (in LMN), is closely aligned with $\hat{\mathbf{k}}$, suggesting the electron crescent is the source of the EBWs.

**Electron crescents drive the EBWs.** Since $\mathbf{B}_M > 0$, the source of the crescents with large phase-space densities at the time of

Fig. 6a should be from the high-density (magnetosheath) side (see Fig. 5a) of the neutral line. The black curves with arrows in Fig. 5b show the Liouville mapping[47] trajectories of 93–631 eV electrons in the plane perpendicular to **B** (more details in the section "Liouville mapping of the electron crescents" and Table 1). Liouville mapping of the measured distribution along $\hat{\mathbf{k}}$ at the time of Fig. 6a (black curve in Fig. 5d) gives the red curve in Fig. 5d. Figure 5e–i presents the detailed comparison of the mapping phase-space densities with the observed ones at different locations (highlighted by the colored bars on the top of Fig. 5c) for 93–631 eV electrons. The good consistency demonstrates that the electron crescents with a positive gradient along $\hat{\mathbf{k}}$ of EBWs are generated by the electron finite gyroradius effect at the electron-scale boundary with a density gradient and a normal electric field.

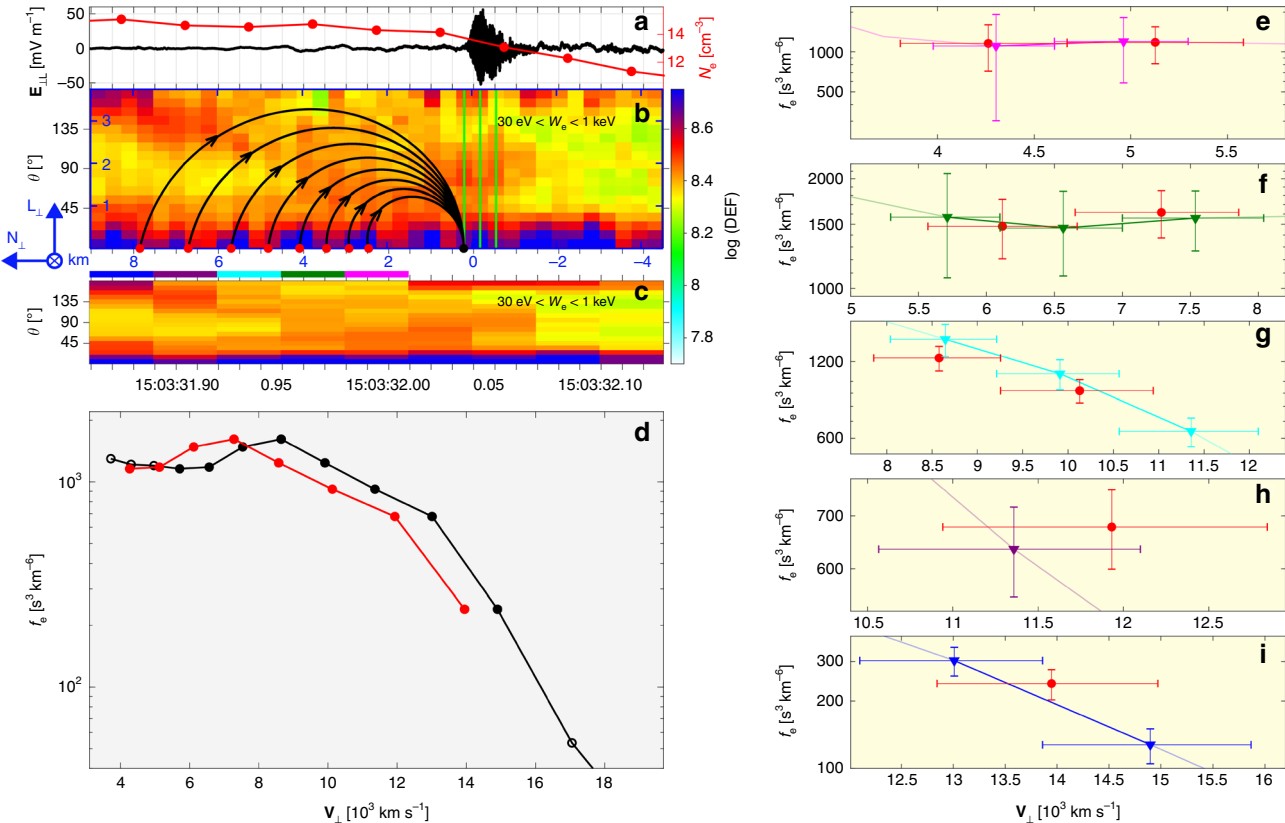

**Fig. 5 Electron crescents generated by finite gyroradius effect. a** $E_{\perp,L}$ with $f > 50$ Hz and $N_e$. **b** Electron pitch-angle spectrogram using 7.5 ms FPI data and trajectories of $93-631$ eV electrons. **c** Electron pitch-angle spectrogram using 30 ms FPI data. In (**d**), the black curve with dots denotes the electron distribution profile along $\hat{k}$ at the time indicated by the first vertical green line in (**b**), and its corresponding Liouville mapping distribution profile is shown by the red curve with dots (see the section "Liouville mapping of the electron crescents" and Table 1). **e–i** Comparison of the mapping distributions (red curve) with five observed ones color-coded on top of (**c**). The errors along velocity are from the energy resolution of FPI, and the electron distribution errors loaded from the MMS data files are from the noise of the particle measurements[25].

We use a fully kinetic dispersion solver[48,49] to verify the generation mechanism of the EBWs. The electron distribution function at the time of Fig. 6a is modeled by a combination of a ring-type distribution[49] for the electron crescents, a parallel-moving Maxwellian, and Maxwellian cores, as shown in Fig. 6g (more details in the section "Electron model distribution function for dispersion relation"). The perpendicular profile (green curve in Fig. 6d) of the model distribution is in good agreement with the observations (black curve in Fig. 6d). Solving the dispersion relation for the model distribution using a kinetic dispersion solver, we find several unstable electron Bernstein modes (see Fig. 6h). The maximum growth rate ($\sim 0.05\ \omega_{ce}$) is found to be along $k_\perp$ direction, pointing towards the perpendicular ring, which models the perpendicular crescent observed by MMS. The phase speed estimated from the maximum growth rates in each of the electron Bernstein branches is 8070 km s$^{-1}$, which is consistent with $V_{ph}$ predicted from the location of the maximum slope in the distribution function (see Fig. 6d). We find similar results by solving a dispersion relation of coupling between electron Bernstein mode and drifting electron beam mode[50,51]. One can note a difference between the MMS observations and the linear theory that the observed EBWs powers peak between the gyro-harmonics, while the peak growth rates locate close to the gyro-harmonics. This difference can possibly be due to the nonlinear effects of the large-amplitude EBWs[51] and using a ring-type distribution instead of a crescent to model the instability.

We estimate the wave electric potential to be $\Phi \sim 13$ V, and the EBWs can trap electrons with perpendicular speed between

$V_{ph} \pm \sqrt{2q_e\Phi/m_e} = (8 \pm 2.1) \times 10^3$ km s$^{-1}$. The electron distributions presented in Fig. 6b, c were observed at the times of the large-amplitude EBWs, and their 1D electron distribution profiles along $\hat{k}$ and $-\hat{k}$ are shown in Fig. 6e, f, respectively. The potential of EBWs is sufficiently large to trap nearly half of the electron crescents and to form the observed plateau in the distributions.

We present an MMS observation of large-amplitude quasi-electrostatic waves at the electron-scale boundary of the Hall current reversal near an EDR encounter. All the properties prove that the quasi-electrostatic waves are EBWs in an over-dense plasma environment. We conclude that the EBWs are driven by the crescent-shaped electron distributions perpendicular to **B**. Here, the local **B** is dominated by the Hall magnetic field embedded in a Hall current reversal. The electron crescents are generated by finite gyroradius effect of the outflow electrons from the magnetosheath side of the neutral line. The EBWs propagate toward the central EDR. The EBWs electric potentials are large enough to thermalize and diffuse the electron crescents near the EDR.

## Discussion

Agyrotropic electron distributions are widely observed in and near EDRs from MMS observations[52]. The agyrotropic electrons may have sufficient free energy to generate different types of intense electrostatic waves. The EBWs reported here and the UH waves in ref. [9] are both driven by the agyrotropic electrons in the plane perpendicular to **B** via wave-mode coupling between beam-

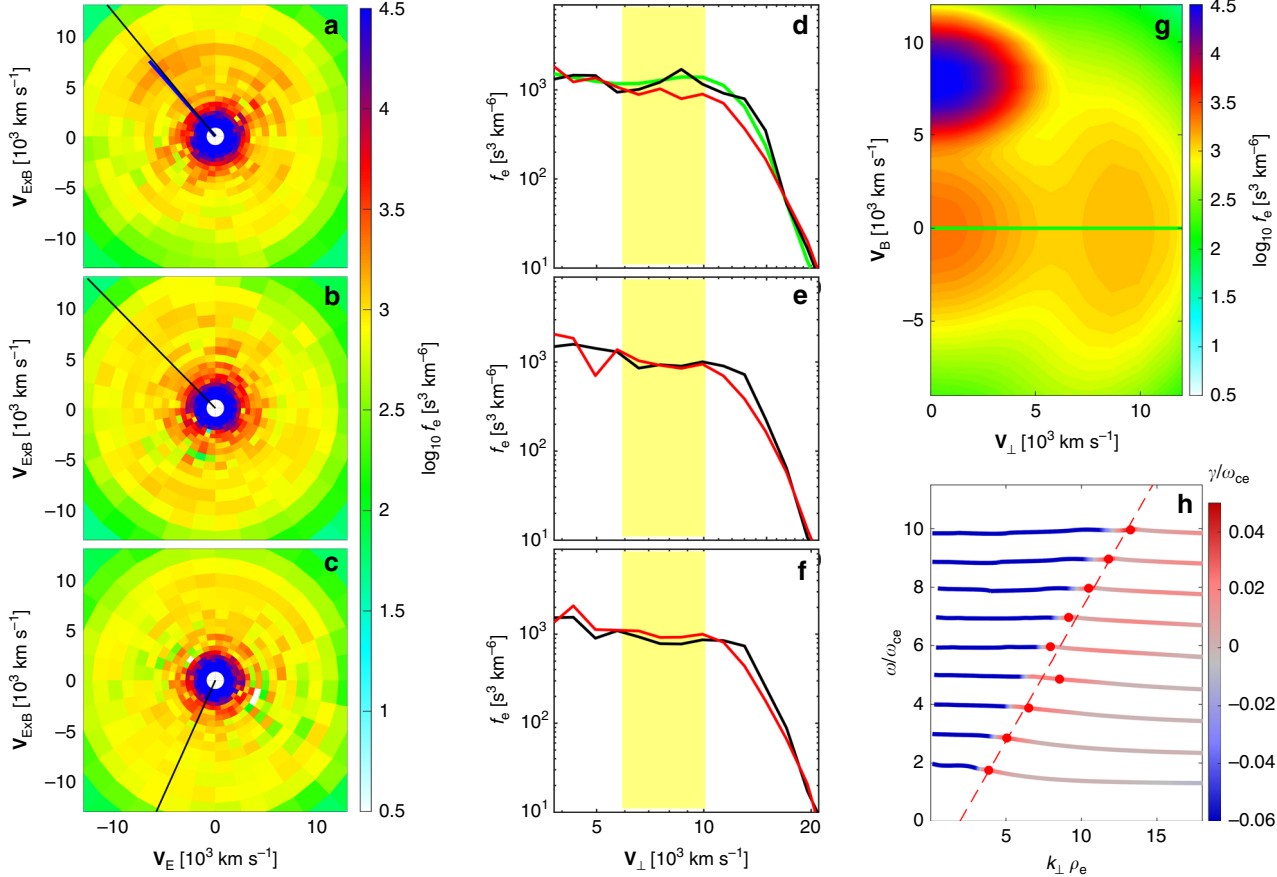

**Fig. 6 Generation mechanism of the EBWs. a–c** Electron distributions at times indicated by the vertical green lines in Fig. 5b. The black lines in each panel denote the $\hat{k}$ direction. The blue line in (**a**) denotes the $\mathbf{E}_{max}$ direction. **d–f** 1D electron distribution profiles along $\hat{k}$ and $-\hat{k}$ from (**a**) to (**c**), respectively. The yellow shading denotes the domain of electron trapping by the EBWs potential. **g** Electron model distribution for the observed distribution in (**a**). The 1D profile along $\mathbf{V}_\perp$ (green line) is presented in (**d**). **h** Dispersion relation of the unstable electron Bernstein mode. The red dots denote the maximum growth rates (fitted by the red-dashed line).

type mode and fundamental wave modes. The two cases are both observed on the magnetosheath side of the reconnection neutral line. The types of the electrostatic waves are probably determined by the specific distribution functions of the agyrotropic electrons and the others. The electron populations in the EBWs case are thermalized (188 eV) electron outflows. The source of the agyrotropic electrons is from the magnetosheath (+N) side, and the crescents have a significant density proportion (~54%) of the total density. The dominant electron populations in the case of the UH waves[9] are magnetosheath (46 eV) electron inflows. The agyrotropic electrons (~5% of the total electrons) gyrate from the −N direction, and may undergo the meandering orbits across the neutral line[8,36]. The EBWs here and the UH waves in ref. [9] are mainly due to the different density proportions of the agyrotropic electrons and the different properties of the background electrons.

The electron agyrotropic distribution functions, that drive the EBWs, are found in the electron outflow near the electron diffusion region. MMS observed unstable distributions just before the EBWs, and diffused distribution while the EBWs were observed. This suggests that the large-amplitude EBWs can change the electron pressure tensor and modify the balance of the reconnection electric field. MMS trajectories of the two events in refs. [8,53] crossed EDRs in a similar way as shown in Fig. 2. We find electrostatic waves with frequencies above the electron cyclotron frequencies at similar electron-scale boundaries of the Hall current reversals. The agyrotropic electron distributions observed at these boundaries are likely to be the source of these

electrostatic waves. The EBWs reported here and the high-frequency electrostatic waves in refs. [8,53] are highly structured at the electron-scale boundaries of the Hall current reversals near EDRs. The cross-field diffusion coefficient of the EBWs is estimated to be $3.6 \times 10^5\ \mathrm{m^2\ s^{-1}}$, using Eqs. (4) and (9) in ref. [54], and the observed wave amplitude of $60\ \mathrm{mV\ m^{-1}}$ and local plasma conditions. The diffusion time is 1.6 s for the electron-scale (0.75 km) boundary shown in Figs. 4 and 5. The high-frequency electrostatic waves reported here and in refs. [8,53] may contribute significantly to the cross-field diffusion of the Hall current reversal boundaries near the electron diffusion regions. The observed magnetic reconnection events are already the results of mixture of all the possible effects. It is difficult to reveal the diffusion effects of the high-frequency waves separately from the data. Further numerical simulations using particle-in-cell models are needed to quantify the systematic effects of the large-amplitude high-frequency waves from linear to nonlinear stages.

## Methods

**Electron velocity distribution functions**. FPI[25] onboard the MMS spacecraft measures the electrons and ions with high time resolution to resolve kinetic-scale plasma dynamic. The burst-mode FPI data provide three-dimensional (3D) electron distribution functions with temporal resolutions of 30 and 7.5 ms[46]. The 32 energy bands of FPI cover electron energies from 10 eV to 30 keV. The angular resolution is 11.25° along both the azimuthal and polar directions. Figure 7a shows the 3D electron distribution functions with $V_e = 8.6 \times 10^3\ \mathrm{km\ s^{-1}}$ of the unstable electron crescents. As shown in Fig. 7b as an example, all the 2D slices used in this study are from the average phase-space densities within ±22.5° from a particular

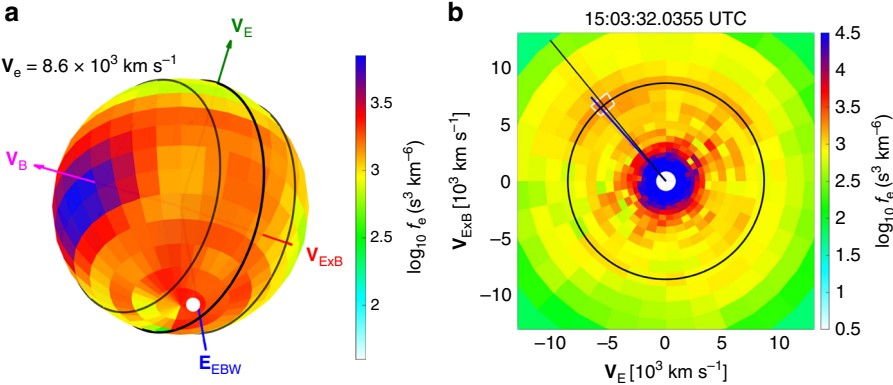

**Fig. 7 Two-dimensional (2D) slice from 3D electron distribution functions. a** Electron distribution functions with electron kinetic velocity of $8.6 \times 10^3$ km s$^{-1}$ (corresponding to 212 eV). The green, magenta, and red arrowed lines denote the directions of $\mathbf{V}_E$, $\mathbf{V}_B$, and $\mathbf{V}_{E \times B}$, respectively. The black circle represents the $\mathbf{V}_E - \mathbf{V}_{E \times B}$ plane perpendicular to the magnetic field **B**, and the two dashed circles show the ranges of $\pm 22.5°$ away from the $\mathbf{V}_E - \mathbf{V}_{E \times B}$ plane. The blue arrowed line labeled as $\mathbf{E}_{EBW}$, show the $\mathbf{E}_{max}$ direction of the EBWs. The white dot in panel (**a**) and the white square in panel (**b**) highlight the location with the largest phase-space density of the electron crescent. The intense phase-space densities close to the $\mathbf{V}_B$ direction corresponds to the parallel magnetosheath electrons moving towards the X line (see Fig. 3f). **b** 2D slice of the distribution function on the $\mathbf{V}_E - \mathbf{V}_{E \times B}$ plane is from the average phase-space densities within $\pm 22.5°$ from the perpendicular plane. The black circle denotes the electron velocity of $8.6 \times 10^3$ km s$^{-1}$. The black and blue lines show the projected directions with the peak phase-space density of the electron crescents and the $\mathbf{E}_{max}$ direction, respectively.

**Table 1 Liouville tracing of the phase-space densities of the crescents in velocity space.**

| $W_e$ [eV] | $V_e$ [$10^3$ km s$^{-1}$] | $d_e$ [km] | $\Delta W_e$ [eV] | $W_e^{Lt}$ [eV] | $V_e^{Lt}$ [$10^3$ km s$^{-1}$] | $\Delta V_e$ [$10^3$ km s$^{-1}$] |
|---|---|---|---|---|---|---|
| 93 | 5.7 | 2.4 | −42 | 51 | 4.2 | −1.5 |
| 122 | 6.6 | 2.9 | −48 | 74 | 5.1 | −1.5 |
| 161 | 7.5 | 3.4 | −56 | 105 | 6.1 | −1.4 |
| 212 | 8.6 | 4.1 | −62 | 150 | 7.2 | −1.4 |
| 279 | 9.9 | 4.8 | −71 | 208 | 8.5 | −1.4 |
| 367 | 11.4 | 5.7 | −80 | 287 | 10.0 | −1.4 |
| 481 | 13.0 | 6.7 | −81 | 400 | 11.8 | −1.2 |
| 631 | 14.9 | 7.8 | −83 | 548 | 13.9 | −1.0 |

$W_e$ and $V_e$ denote the electron energy and kinetic velocity, respectively. $d_e$ is the gyro-diameter (see Fig. 5b). $\Delta W_e$ and $\Delta V_e$ represent the changes of the electron energy and velocity across the gyro-diameter (see Fig. 5d), while the superscript Lt denotes the Liouville tracing values

plane, while unit vectors are projected when they are within $\pm 22.5°$ respect to the plane.

**Liouville mapping of the electron crescents**. The crescents generating the large-amplitude electron Bernstein waves gyrate from the magnetosheath (+N) side of the neutral line. We get the mapped electron phase-space density by using the Liouville's theorem[47] that the electron phase-space density are conserved along the particle trajectories throughout the gyro-motion at the electron-scale boundary (see Fig. 5). We assume one-dimensional variation of the magnetic field **B** and the electric field **E** around the EBWs observations. Here, **B** is dominated by $\mathbf{B}_M$. The electric potential of the electron increases when tracing towards the high-density region[55] (see Fig. 5a, b, d). The electron trajectories (see Fig. 5b) in the plane perpendicular to **B** are obtained by solving the electron motion equation based on observed electric and magnetic fields. Table 1 shows the details of the electron distribution function tracing in velocity space, while the curves in Fig. 5e–i also consider the energy resolution and errors of the phase-space densities.

**Electron model distribution function for dispersion relation**. We use a gyrotropic ring-type distribution function to model the observed crescents, while the rest of the observed electron populations is modeled by one drifting (along the field line) Maxwellian and two nondrifting Maxwellian cores. The background magnetic field is 30 nT, and the electron total number density is 14.1 cm$^{-3}$. The gyrotropic ring is from superposition of multiple ring-type distribution functions[49]:

$$f_r(V_\parallel, V_\perp) = \frac{N_r}{\pi^{3/2} \delta v_\perp^2 \delta v_\parallel \Gamma} \exp\left[-\frac{(V_\perp - v_{\perp r})^2}{\delta v_\perp^2}\right] \exp\left[-\frac{(V_\parallel - v_{\parallel r})^2}{\delta v_\parallel^2}\right], \quad (1)$$

$$\Gamma = \exp\left(-\frac{v_{\perp r}^2}{\delta v_\perp^2}\right) + \frac{\sqrt{\pi} v_{\perp r}}{\delta v_\perp} \text{erfc}\left[-\frac{v_{\perp r}}{\delta v_\perp}\right], \quad (2)$$

where erfc is the complementary error function. The parameters, number density ($N_r$, in cm$^{-3}$), parallel and perpendicular thermal speeds ($\delta v_\parallel$ and $\delta v_\perp$,

in km s$^{-1}$), the parallel speed ($v_{\parallel r}$, in km s$^{-1}$), and the perpendicular ring speed ($v_{\perp r}$, in km s$^{-1}$), are listed in Supplementary Data 1. The Maxwellian cores contain 65 eV electrons with a number density of 1.35 cm$^{-3}$ and 320 eV electrons with a number density of 2.02 cm$^{-3}$. The parallel-moving Maxwellian electrons have a number density of 3.38 cm$^{-3}$, a parallel speed of 8000 km s$^{-1}$, perpendicular temperature of 20 eV and parallel temperature of 10 eV. All the parameters are adjusted to have a best fitting of the crescents and a relatively good fitting of the rest.

## Data availability
MMS L2 data are available from the MMS Science Data Center (https://lasp.colorado.edu/mms/sdc/public).

## Code availability
All of the data plots in this study are generated with the IRFU-Matlab software applied to the publicly available MMS database. The IRFU-Matlab software is available by downloading from https://github.com/irfu/irfu-matlab.

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

## Acknowledgements

We thank the MMS team and instrument principal investigators for data access and support. W.Y.L. appreciates the helpful discussion with Dr. B. Lembège from LATMOS Laboratoire, France and Dr. Lynn B. Wilson III from Goddard Space Flight Center, USA. This work was supported by the NNSFC Grant Nos. 41504114, 41731070, 41574159 and 41974170, Swedish National Space Board, Grant Nos. 164/14, 128/17 and 176/15, Swedish Research Council 2016-05507, Strategic Pioneer Program on Space Science of Chinese Academy of Sciences, Grant No. XDA15052500, and in part by the Specialized Research Fund for State Key Laboratories of China. W.Y.L. is also supported by the Youth Innovation Promotion Association (2018177) and opening fund of State Key Laboratory of Lunar and Planetary Sciences (Macau FDCT Grand No. 119/2017/A3). S.T.-R. acknowledges support of the of the Ministry of Economy and Competitiveness (MINECO) of Spain (Grant FIS2017-90102-R). The work at IRAP and the French involvement (SCM) on MMS were supported by CNES and CNRS.

## Author contributions

W.Y.L. and D.B.G. carried out the data analysis, interpretation and manuscript preparation. Y.V.K., A.V., M.A., B.B.T., C.W., K.F., C.N. and S.T.-R. contributed to the data interpretation and manuscript preparation. K.M. and K.L. performed the wave analysis and contributed to the data interpretation. P.-A.L., Y.V.K., D.B.G., R.E.E. and R.B.T. contributed to the development and operation of the electric-field measurements. A.C.R., J.C.D., D.J.G., B.L.G. and B.L. contributed to the development and operation of the Fast Plasma instruments. F.P., W.M., O.L.C. and C.T.R. contributed to the development and operation of the magnetic-field measurements. J.L.B. led the design and operation of the MMS mission.

## Competing interests

The authors declare no competing interests.
