## [Peer Review File · Nature Communications]

Reviewers' comments:

Reviewer #1 (Remarks to the Author):

This article is dense and the figures are dense. Nonetheless it convincingly makes its point about the origin and magnitude, hence consequences, of the EBW in the current reversal region.

The authors can make a lot of improvements to the presentation to make the article more accessible. Some of the comments below relate to that. There are a few discrepancies in the descriptions of certain phenomena, and a few places where essential information is omitted. Others of the comments below relate to those.

Comments on the figures:

Remove as many traces from line plots, leaving only the essential ones needed to make your argument. In many cases all three components of B, J, and V are shown when only one, usually the L-component, is discussed in the text. The extra traces are not an advantage and obscure the main point.

Remove unnecessary lines from the electron distribution plots. Remove any not needed for the argumentation, such as, for example, in Fig. 4e.gi do not show both k and -k directions, one of these suffices.

Separate Fig. 3a as its own figure. First of all, it's referred to much earlier in the text than Fig. 3. Second, being printed so small makes it misleading. For example, Fig. 3a suggests that neutral line and current boundary should coincide, but this is not true in the case study. It would be nice if a larger version of the Figure could show this.

Show zero as a dotted line in the cases where zero-crossings are important (if unnecessary traces have been removed, this will be possible).

Include the vertical red and black lines indicating neutral sheet and separatrix in Fig. 2, to better connect it to Fig. 1.

In Fig. 4d, light colored traces do not appear to be discussed in text and should be removed. The mini-panels and associated color-coding should be described in the caption.

Comments on the text:

Last sentence of abstract is unclear: what is meant by "from EDR to IDR"? The observations are in the EDR, what can they say about the IDR? The discussion section p. 11-12 does not seem to address anything related to this phrasing in the abstract.

On p. 4, Fig 4a is referred to, where Fig 3a is meant, but I urge breaking that into its own separate Figure.

Bottom p. 4, indicate that the lower hybrid waves are "not shown" (unless they are---are they evident in the Delta-E panel i? If so, point this out.)

Middle of p. 6, current obtained from plasma data. Some discussion is needed here of the energy range of the plasma data, and whether it captures all or most of the current, and how you know that.

Bottom p. 6, suggest adding "which are the Hall current peaks" after "respectively", to explicitly tell the reader the physical significance of the yellow bars.

p 7, caption for (d), it appears some kind of low-pass filter has been applied? (otherwise, the wave field shown in Fig 3, nearly 50 mV/m, would dominate?) Add discussion of any applied filtering.

p 7 4 lines from bottom, text refers to beams as parallel to B, figure shows M. In text, remind reader B is (mostly) along M.

p 8, and Fig. 3: here, I assume some kind of high-pass filter has been applied to E, hence it's called "high-frequency E." Please give filter details.

Fig. 3 and text: Explain how the yellow region is defined. Is it strictly on wave power or something else? Is it arbitrary "by eye" or please give a quantitative criterion defining it.

State in text or caption what exact time interval was used for the spectral/MVA analyses in Fig. 3 (e.g., was it the yellow region?)

There is some significant variation of Nf_{ce} across the yellow region in Fig. 3f. Tell in text how the specific values of Nf_{ce} indicated by dashed lines in Fig 3g were determined.

Top of p 8: text refers to perp drift as $E \times B / B^2$, Fig. 3 refers to same drift as V. Help the reader out here, use the same nomenclature in both places or put $V_{perp} = E \times B / B^2$ in the text.

Same place: $E_{perp} \gg E_{par}$ appears to hold only in the yellow region, text implies everywhere.

Middle p 8 "which is close to the reconnecting field direction L" indicate not shown (unless it is?) Tell how close it is.

Top of p 9: There seems to be a discrepancy between text and figure caption concerning the 1-d electron distribution cuts shown in Fig. 3f (and presumably also those in 3hj). Figure indicates it's the k-direction, text indicates it's the direction to the centroid of the crescent. If the latter, more information needs to be given about how this centroid was determined.

Starting p. 9, the direction k starts being referred to. Presumably this means the direction of the wave propagation? It needs to be defined and told how it was determined (from MVA perhaps?).

On p. 9 toward bottom, Where Liouville mapping trajectories are introduced, a better explanation needs to be given of what these are. Reference should be made here to the appendix, so the reader knows it exists.

It is unclear to me what is meant by "the non-uniform shifting of the phase space densities." Does this have something to do with the relation of the red curve to the black curve in Fig. 4d?

Middle p. 10 "...along a phase velocity direction pointing towards the crescents." Indicate this is not shown (unless it is? in which case indicate how it is shown.)

Middle p. 10 and Fig. 4l: Fig. 4l appears to show max growth rate of the EBW very close to the cyclotron harmonics, whereas in the data, the wave power peaks in between the harmonics (Fig. 3g). Some discussion is needed to assess this and whether this is a significant discrepancy between theory and experiment.

Bottom p. 10 "trap a large fraction: It doesn't seem too hard to give the approximate fraction (half? quarter? three quarters?)

Reviewer #2 (Remarks to the Author):

This paper reported an interesting magnetic reconnection electron diffusion region (EDR) encounter case by the Magnetospheric Multiscale spacecraft (MMS), in which they found large-amplitude electron Bernstein waves. The results are very important for the space physics community as well as fusion plasma, and the paper is well organized. Therefore, I support the paper to be published in its current form. Nevertheless, I have some minor and (optional) suggestions which may help for the paper.

1. It may be better to include a bit more introduction about electron Bernstein waves, which could bring a broader interest.
2. Page 6. it may be better to provide an explanation why the width of Hall current is much smaller than the ion inertia scale?
3. Page 6. line 3 of the last paragraph, density \rightarrow phase density

Reviewer #3 (Remarks to the Author):

NATURE PAPER LI ET AL, "ELECTRON BERNSTEIN WAVES..."

This paper is on an interesting topic that potentially makes a contribution to the relative importance of wave-particle vs finite Larmor radius effects in enabling magnetic reconnection in nearly collisionless plasmas. The authors do not marshal their arguments well from the beautiful measurements. Many figures are presented as having self evident properties. Others reveal internally contradictory properties.

As an example: Pitch angle pictures are shown as relief maps implying two independent variables with the 3rd the color coding of intensity, and then refers to them as showing agyrotropy. A typical pitch

angle portrait of this type suppresses gyrophase of the observed particles to make this plot one of energy vs $\text{acos}(V \cdot B / |V| / |B|)$; such a plot by definition cannot reveal agyrotropy!

If it is a map vs gyro phase, then some other variable in velocity space is being suppressed, since to show that particles of common pitch angle in the fluid rest frame must be contrasted in intensity

vs gyro phase suppressing all other pitch angles. Unfortunately the authors have clearly indicated what this figure shows and or how it does so.

Otherwise Figure 2f-h is about, the authors need to state that clearly, including discussion how the other parts of velocity space were suppressed.

Later it was stated that the locales of the observations reported were where crescent distributions were observed. Since it is the apparent thesis of this paper that the crescents are the cause of the electron Bernstein waves it is reasonable for the reader to be shown data that (a) show the crescents

rather than referring to their existence, and (b) show the persistence of crescents across the domain of the radiation, with a demonstrable plottable measure of crescent presence. Currently an interval is

simply declared to be region where agyrotropic distributions are seen. There are a number of scalar measures of agyrotropy that could be used to document what interval in the data quantitatively reflect

the stated properties that at present are declared as a fact. The author's seem to be unaware that recording such agyrotropy in conjunction with other magnetic geometry indicators may already be enough to all reconnection to proceed. There seems to be interest only in the waves for this purpose without considering their role as causal to reconnection or reflexive of the gyro-mechanics that caused

the agyrotropy to form in the beginning. This situation is complicated by the heavy aliasing of the plasma measurements by the time variations of the waves being reported. This heavy aliasing also clouds the instability analysis reported to explain the EBW.

The preparation for the dispersion analysis also appears sketchy to this reviewer. Shouldn't the sufficiency of the eVDF for this modeling be judged based on reproducing the entire measured distribution

function before doing an initial value instability analysis? The present text appeals to having matched the perpendicular cut of the distribution as a sufficient test for the initial value simulation. Shouldn't the modeled input crescent distribution for instability analysis exhibit its faithfulness by

plotting f_{disp} vs f_{obs} as a Y vs X scatter plot on log, log paper using every velocity pixel measured to demonstrate the realism of the input for the analysis for the dispersion solver.

What is the dispersion solver used? It should be referenced.

Other places: undescribed pitch angles pictures contain overplotted 3 orthogonal L,M,N directions superposed on the 2 independent variables to draw conclusions about who is pointed in which directions.

Flat topped electron distributions in the magnetosheath are common; the paper interprets the flattening of the distribution as evidence for the role of EBW in regulating the reconnection process. Because the initial value distribution function is not shown, there is hardly a before and after demonstration that the diffusion process was important.

Isn't the 7.5ms FPI electron observations heavily aliased in the EBW interval of 16ms? If so how is there adequate information for showing that it is unstable using initial value instability measures? Clearly the waves are large amplitude, but the argument about their being important for formation of the diffusion region or field line erosion escapes this reviewer. The instability analysis attempted presupposes the agyrotropic distribution is created first and the EBW evolves from it. Isn't the formation of the agyrotropic distribution of this type already capable of disrupting frozen flux conservation? To advance the argument it would appear necessary to comment on the relative importance of Div P effects seen and those of these waves. From the abstract one gets the impression that the paper had assayed this competition; disappointingly it has not.

p 9 typescript: "The crescent shaped electron distribution [not shown] is close to the ExB direction" This is difficult to swallow since the distribution lives in 3-dimension velocity space and is sub sonic, so in what sense is this 4 surface close to a direction?

Recommendation: The 25 authors can surely marshal the experimental argument more cogently than in this submitted manuscript. To the principal author: give the submitted manuscript to 3 subgroups of authors for comment about its clarity. Take their comments seriously and revise to alleviate issues like those above.

This is an interesting topic, but that does not mean its is publishable in its present form. The exposition of the paper's experimental data seems less conversant with clear description of the necessary plasma supporting information. The graphics for, and diagnosis of the

crescent distribution is absent or presently unintelligible. Seems unlikely that authors Rager- Giles on the authorship list have approved the submitted article. The article's claim for importance in the electron diffusion region has not separated its role as progenitor or reactor to the agyrotropy, contrary to the impression of the last line in the abstract.

We appreciate all the comments and suggestions from the three reviewers, and have addressed all the points and revised the manuscript. The detailed answers are listed below.

Reviewers #1' comments:

This article is dense and the figures are dense. Nonetheless it convincingly makes its point about the origin and magnitude, hence consequences, of the EBW in the current reversal region. The authors can make a lot of improvements to the presentation to make the article more accessible. Some of the comments below relate to that. There are a few discrepancies in the descriptions of certain phenomena, and a few places where essential information is omitted.

Reply: We thank the reviewer for the comments and suggestions. The figures are modified to highlight the key points of this study. The text is revised to clarify some ambiguous expression and a few discrepancies and fix some typos. The following part contains the detailed reply to each comment.

Comments on the figures:

1. Remove as many traces from line plots, leaving only the essential ones needed to make your argument. In many cases all three components of B, J, and V are shown when only one, usually the L-component, is discussed in the text. The extra traces are not an advantage and obscure the main point. Remove unnecessary lines from the electron distribution plots. Remove any not needed for the argumentation, such as, for example, in Fig. 4egi do not show both k and -k directions, one of these suffices. Separate Fig. 3a as its own figure. First of all, it's referred to much earlier in the text than Fig. 3. Second, being printed so small makes it misleading. For example, Fig. 3a suggests that neutral line and current boundary should coincide, but this is not true in the case study. It would be nice if a larger version of the Figure could show this. Show zero as a dotted line in the cases where zero-crossings are important (if unnecessary traces have been removed, this will be possible). Include the vertical red and black lines indicating neutral sheet and separatrix in Fig. 2, to better connect it to Fig. 1. In Fig. 4d, light colored traces do not appear to be discussed in text and should be removed. The mini-panels and associated color-coding should be described in the caption.

Reply: In the re-submission, all figures are modified. We removed several traces and lines from Figures 1, 2, 3 and 4 of the previous submission. The sketch of the MMS crossing was modified and separated as its own figure, Figure 2 in the revision. The vertical red and black lines are included in Figure 3 to indicate the neutral sheet and the magnetospheric separatrix locations. We revised the caption of Figure 5 and add Table 1 to clarify the light colored traces in the mini-panels and the Liouville mapping.

2. Last sentence of abstract is unclear: what is meant by "from EDR to IDR"? The observations are in the EDR, what can they say about the IDR? The discussion section p. 11-12 does not seem to address anything related to this phrasing in the abstract.

Reply: We revise this sentence to be “The EBWs contribute to the cross-field diffusion of the electron-scale boundary of the Hall current reversal near the electron diffusion region”.

3. On p. 4, Fig 4a is referred to, where Fig 3a is meant, but I urge breaking that into its own separate Figure.

Reply: The sketch is modified and separated as Figure 2 in the updated version.

4. Bottom p. 4, indicate that the lower hybrid waves are "not shown" (unless they are---are they evident in the Delta-E panel i? If so, point this out.)

Reply: The major argument of this study is the high-frequency electron Bernstein waves on the magnetosheath side of the neutral line. To focus on this point, we remove this sentence about the lower-hybrid waves.

5. Middle of p. 6, current obtained from plasma data. Some discussion is needed here of the energy range of the plasma data, and whether it captures all or most of the current, and how you know that.

Reply: The energy ranges of FPI ion and electron detectors are from 10 eV to 30 keV, which can provide a good energy coverage of the plasmas at the magnetopause. Also, the four dual-spectrometers for electrons and ions accomplish high-temporal resolutions of the plasma detections. An aspect of such improvement of plasma instruments is that we can calculate electric currents $\mathbf{J} = e(n_i \mathbf{V}_i - n_e \mathbf{V}_e)$ directly from particle data [e.g., *Phan et al., 2016; Graham et al., 2016*]. As shown in Figure R1, the electric currents from FPI agree well with those from the curlometer method [*Dunlop et al., 1988*] using \mathbf{B} from four spacecraft. We revised the caption of Figure 3 and add two references on this point.

Figure R1. Comparison of \mathbf{J} from FPI particle and magnetic field data. (a) \mathbf{B} . (b) ion and (c) electron energy fluxes. (d)-(f) electric current estimated from the particle data and the curlometer method in the LMN coordinates.

6. Bottom p. 6, suggest adding "which are the Hall current peaks" after "respectively", to explicitly tell the reader the physical significance of the yellow bars.

Reply: Revised as been suggested.

7. p 7, caption for (d), it appears some kind of low-pass filter has been applied? (otherwise, the wave field shown in Fig 3, nearly 50 mV/m, would dominate?) Add discussion of any applied filtering.

Reply: The filtering information of the \mathbf{E} data is added in the captions and the figures.

8. p 7 4 lines from bottom, text refers to beams as parallel to \mathbf{B} , figure shows \mathbf{M} . In text, remind reader \mathbf{B} is (mostly) along \mathbf{M} .

Reply: This point is highlighted in the re-submission.

9. p 8, and Fig. 3: here, I assume some kind of high-pass filter has been applied to \mathbf{E} , hence it's called "high-frequency \mathbf{E} ." Please give filter details.

Reply: The filtering information of the \mathbf{E} data is added in the captions and the figures.

10. Fig. 3 and text: Explain how the yellow region is defined. Is it strictly on wave power or something else? Is it arbitrary "by eye" or please give a quantitative criterion defining it. State in text or caption what exact time interval was used for the spectral/MVA analyses in Fig. 3 (e.g., was it the yellow region?)

Reply: As shown in Fig. R2a, the yellow-shaded region is defined by $|\mathbf{E}_{\perp\perp}| > E_{\max}/e^2$, where E_{\max} is maximum of the $|\mathbf{E}_{\perp\perp}|$ fluctuations. The exact time interval (15:03:32.037 UT to 15:03:32.054 UT) is stated in the caption of Figure 4 of the new submission.

Figure R2: Electron Bernstein waves observed by MMS1. (a) $|E_{\perp L}|$, (b) Spectrogram of $|E_{\perp L}|$, (c) electron cyclotron frequency f_{ce} . The red dot in (a) denotes the maximum of the fluctuating $|E_{\perp L}|$ (E_{max}), and the horizontal red dashed line represents E_{max}/e^2 , where $e \sim 2.718$ is the Euler identity. (d) Power spectrum of $|E_{\perp}|$ within the yellow-shaded region. (e)-(g) Power spectrums of the three sub-region of the yellow-shaded region. The vertical lines in (d)-(g) denote the average f_{ce} within each interval and their harmonics, and the cyan bars show their standard deviations.

11. There is some significant variation of Nf_{ce} across the yellow region in Fig. 3ef. Tell in text how the specific values of Nf_{ce} indicated by dashed lines in Fig 3g were determined.

Reply: Figure R2c shows the variation of f_{ce} during the observations of the electron Bernstein waves. The $f_{ce} \sim 857$ Hz is estimated from the average value of the f_{ce} within the interval of the yellow-shaded region, with a standard deviation of 9.9 Hz. Figure R2d shows the power spectrum of $|E_{\perp}|$ within the yellow region, with f_{ce} and its harmonics. The standard deviations are shown by the cyan bars. In the yellow-shaded region, we find distinct spectral peaks separated in frequency by approximately the electron cyclotron frequency. In the re-submission, we revised the caption of Figure 4 (previously Fig. 3) to clarify this point.

12. Top of p 8: text refers to perp drift as ExB/B^2 , Fig. 3 refers to same drift as V . Help the reader out here, use the same nomenclature in both places or put $V_{\perp} = ExB/B^2$ in the text. Same place: $E_{\perp} \gg E_{\parallel}$ appears to hold only in the yellow region, text implies everywhere. Middle p 8 "which is close to the reconnecting field direction L " indicate not shown (unless it is?) Tell how close it is.

Reply: The text is revised as '... the point where $V_{\perp} = \mathbf{E} \times \mathbf{B} / B^2$ changes sign ... In the yellow-shaded region, we can see that $E_{\perp} \gg E_{\parallel}$, ...'. The y axis title of Fig. 4c is revised to be V_{\perp} . In the yellow-shaded region, the angle between $E_{\perp L}$ and L is about 19° . The text is also revised.

13. Starting p. 9, the direction k starts being referred to. Presumably this means the direction of the wave propagation? It needs to be defined and told how it was determined (from MVA perhaps?).

Reply: $\hat{\mathbf{k}} = \frac{\mathbf{k}}{|\mathbf{k}|}$ is the direction of the phase velocity, where \mathbf{k} is the wave vector. It is determined by the direction with the largest crescent phase-space density in velocity space (see more details in Fig. R3 and the following reply), which is 7.3° away from the maximum variation direction of electric field fluctuations E_{max} . The angle is smaller than the angular resolution of FPI (11.25°). The text is revised to clarify this point.

14. Top of p 9: There seems to be a discrepancy between text and figure caption concerning the 1-d electron distribution cuts shown in Fig. 3f (and presumably also those in 3hj). Figure indicates it's the k -direction, text indicates it's the direction to the centroid of the crescent. If the latter, more information needs to be given about how this centroid was determined.

Reply: Figure R3 shows how the 2D slices (Fig. 5e of the re-submission) of the 3D distribution functions are suppressed. The white dot in Fig. R3a denotes the largest phase-space density of

the electron crescent, and the direction of wave vector \mathbf{k} is determined by the location of the white dot in velocity space. The 1D electron distribution cuts are taken from the \mathbf{k} direction, which is 7.3° away from the \mathbf{E}_{\max} direction. The text is revised, and one subsection with Fig. R3 is added in the *Method* section to answer several questions on the electron distribution functions.

Figure R3: 7.5 ms electron distribution function slice. (a) Skymap of the electron distribution functions with electron velocity of 8.6×10^3 km/s (corresponding to 212 eV). The green, magenta, and red arrowed lines denote the directions of \mathbf{V}_E , \mathbf{V}_B , and \mathbf{V}_{ExB} , respectively. The black circle represents the \mathbf{V}_E - \mathbf{V}_{ExB} plane perpendicular to the magnetic field \mathbf{B} , and the two dashed circles show the ranges of $\pm 22.5^\circ$ away from the \mathbf{V}_E - \mathbf{V}_{ExB} plane. The blue arrowed line show the \mathbf{E}_{\max} direction of the EBWs, and the white dot highlights the location with the largest phase-space density of the electron crescent. The intense phase-space densities close to the \mathbf{V}_B direction corresponds to the parallel magnetosheath electrons moving towards the X line. The 2D slice of the distribution function in the \mathbf{V}_E - \mathbf{V}_{ExB} plane (b) is from the averaging of the phase-space densities within $\pm 22.5^\circ$ from the perpendicular plane. The black circle in (b) denotes the average phase-space densities from (a). The black and blue lines shows the projected directions with the peak phase-space density of the crescent and \mathbf{E}_{\max} .

15. On p. 9 toward bottom, Where Liouville mapping trajectories are introduced, a better explanation needs to be given of what these are. Reference should be made here to the appendix, so the reader knows it exists. It is unclear to me what is meant by "the non-uniform shifting of the phase space densities." Does this have something to do with the relation of the red curve to the black curve in Fig. 4d?

Reply: In the re-submission, a table is added in the *Liouville mapping of the electron crescents* subsection to show the details of the Liouville mapping trajectories. The exact values of the Liouville tracing in velocity space are presented in the last column of the table to show "the non-uniform shifting of the phase space densities". The light colored traces in Fig. 5(d) is removed, and the text and the caption of Fig. 5 are both revised.

16. Middle p. 10 "...along a phase velocity direction pointing towards the crescents." Indicate this is not shown (unless it is? in which case indicate how it is shown.)

Reply: Figure R4 shows the dispersion surface of the 4th harmonic of the electron Bernstein mode from our wave analysis using the dispersion solver of *Min and Liu* [2015, 2016]. The region

with positive growth rates (red part) is close to the exact perpendicular direction \mathbf{k}_\perp . The direction with the largest growth rate is about 0.8° away from \mathbf{k}_\perp , which points towards the ring-type distribution (modelling the crescent). This sentence is revised in the re-submission.

Figure R4: Dispersion relation of the 4th harmonic ($4 f_{ce} < f < 5 f_{ce}$) of the electron Bernstein mode.

17. Middle p. 10 and Fig. 4l: Fig. 4l appears to show max growth rate of the EBW very close to the cyclotron harmonics, whereas in the data, the wave power peaks in between the harmonics (Fig. 3g). Some discussion is needed to assess this and whether this is a significant discrepancy between theory and experiment.

Reply: The dispersion surfaces from the linear solver show that the maximum growth rates are close to the cyclotron harmonics, and the growth rates of different harmonic branches are similar. MMS observations show that the wave powers peak in between the harmonics. Also the largest wave power locates in the higher harmonic in the earlier time and in the lower harmonics in the later time (Figure R3e-g), with relatively stronger magnetic field fluctuations (Fig. 4e of the re-submission). Those are probably due to the nonlinear effect of the large-amplitude EBWs. *Muschietti and Lembege* [2013] investigated the linear-nonlinear evolution of the electron Bernstein waves driven by the electron cyclotron drift instability (ECDI) from the ion beam versus the electrons. In their study, the resonance broadening and the ion trapping effect during the nonlinear stage are responsible for the frequency and power variations. In our case, a kinetic simulation based on initial value conditions can show the evolution and reveal the details of the nonlinear effect, and K. Liu (one of the co-authors) is considering such a PIC simulation.

Besides, the electron crescent is currently modelled by a ring-type gyrotropic distribution function. This could also be responsible for the discrepancy. Recently, we find a fully new kinetic dispersion relation solver developed by Dr. Xie (<https://github.com/hsxie/pdrk/>), which can include the perpendicular drift velocity of the Maxwellian distributions to analyze magnetized plasma waves. With this solver, we plan to use drifting Maxwellian to model the crescent; thus, confirm whether the discrepancy is from the model distribution functions.

18. Bottom p. 10 "trap a large fraction: It doesn't seem too hard to give the approximate fraction (half? quarter? three quarters?)

Reply: Nearly half of the crescent distribution is trapped, and the text is revised.

References:

1. Dunlop, M. W. et al. (1988), Analysis of multipoint magnetometer data, *Adv. Space Res.*, 8, 273.
2. Graham, D. B., et al. (2016), Electron currents and heating in the ion diffusion region of asymmetric reconnection, *Geophys. Res. Lett.*, 43, 4691–4700.
3. Min, K. & Liu, K. (2015) Fast magnetosonic waves driven by shell velocity distributions. *J. Geophys. Res. Space Physics* 120, 2739–2753.
4. Min, K. & Liu, K. (2016) Understanding the growth rate patterns of ion Bernstein instabilities driven by ring-like proton velocity distributions. *J. Geophys. Res. Space Physics* 121, 3036–3049.
5. Muschietti, L. & Lembège, B. (2013) Microturbulence in the electron cyclotron frequency range at perpendicular supercritical shocks. *J. Geophys. Res. Space Physics* 118, 2267–2285.
6. Phan, T. D. et al. (2016) MMS observations of electron-scale filamentary currents in the reconnection exhaust and near the x line. *Geophys. Res. Lett.* 43, 6060–6069.

Reviewer #2 (Remarks to the Author):

This paper reported an interesting magnetic reconnection electron diffusion region (EDR) encounter case by the Magnetospheric Multiscale spacecraft (MMS), in which they found large-amplitude electron Bernstein waves. The results is very important for space physics community as well as fusion plasma, and the paper is well organized. Therefore, I support to paper to be published as current form. Nevertheless, I have some minor and (optional) suggestions which may help for the paper.

Reply: We thank the reviewer for the comments and suggestions. We include more introduction of the electron Bernstein waves studies of the general plasma community. The manuscript is revised to clarify some ambiguous expression and a few discrepancies.

1. It may be better to include a bit more introduction about electron Bernstein waves, which could bring a broad interest.

Reply: The electron Bernstein waves have been observed around Earth's bow shock [Wilson *et al.*, 2010] and inside the magnetosphere [Lou *et al.*, 2018], and the EBWs are also adopted for efficient heating of plasmas in fusion devices [Laqua *et al.*, 2007]. We include those introduction about the EBWs in the re-submission.

2. Page 6. it maybe better to provide an explanation why the width of Hall current is much smaller than the ion inertia scale?

Reply: In magnetic reconnection, the Hall effect including the Hall current is due to the charge separation between the ions and the electrons, and can be observed in a scale smaller than the typical ion inertial length. Much stronger charge separation close to the X line drives stronger Hall effect with a spatial scale close to electron inertial scale. The electron-scale Hall effect has been widely analyzed by the particle-in-cell (PIC) simulations [e.g., Shay *et al.*, 2016], and MMS brings an unique opportunity to reveal those effects in space plasmas [e.g., Wang *et al.*, 2017].

3. Page 6. line 3 of the last paragraph, density -> phase density

Reply: Fixed.

References:

1. Laqua, H. P. (2007), Electron Bernstein wave heating and diagnostic, *Plasma Physics and Controlled Fusion*, 49, R1–R42.
2. Lou, Y. et al. (2018), Statistical distributions of dayside ECH waves observed by mms. *Geophys. Res. Lett.*, 45, 12,730–12,738.
3. Shay, M. A. et al. (2016), Kinetic signatures of the region surrounding the X line in asymmetric (magnetopause) reconnection, *Geophys. Res. Lett.*, 43, 4145–4154.
4. Wang, R., et al. (2017), Electron-Scale Quadrants of the Hall Magnetic Field Observed by the Magnetospheric Multiscale spacecraft during Asymmetric Reconnection, *Physical Review Letters*, 118(17), 175101.

5. Wilson III, L. B. et al. (2010), Large-amplitude electrostatic waves observed at a supercritical inter-planetary shock, *J. Geophys. Res. Space Physics*, 115.

Reviewer #3 (Remarks to the Author):

This paper is on an interesting topic that potentially makes a contribution to the relative importance of wave-particle vs finite Larmor radius effects in enabling magnetic reconnection in nearly collisionless plasmas. The authors do not marshal their arguments well from the beautiful measurements. Many figures are presented as having self evident properties. Others reveal internally contradictory properties.

Reply: We thank the reviewer for the comments and suggestions. The manuscript is revised to present a clearer argument of the electron Bernstein waves near the electron diffusion region. A few discrepancies in the presentation and some typos are also fixed. The following part contains the detailed reply to each comment.

1. As an example: Pitch angle pictures are shown as relief maps implying two independent variables with the 3rd the color coding of intensity, and then refers to them as showing agyrotropy. A typical pitch angle portrait of this type suppresses gyrophase of the observed particles to make this plot one of energy vs $\cos(\theta) = V_{\parallel}/|V|$; such a plot by definition cannot reveal agyrotropy! If it is a map vs gyro phase, then some other variable in velocity space is being suppressed, since to show that particles of common pitch angle in the fluid rest frame must be contrasted in intensity vs gyro phase suppressing all other pitch angles. Unfortunately the authors have clearly indicated what this figure shows and or how it does so. Otherwise Figure 2f-h is about, the authors need to state that clearly, including discussion how the other parts of velocity space were suppressed. Later it was stated that the locales of the observations reported were where crescent distributions were observed. Since it is the apparent thesis of this paper that the crescents are the cause of the electron Bernstein waves it is reasonable for the reader to be shown data that (a) show the crescents rather than referring to their existence, and (b) show the persistence of crescents across the domain of the radiation, with a demonstrable plottable measure of crescent presence. Currently an interval is simply declared to be region where agyrotropic distributions are seen. There are a number of scalar measures of agyrotropy that could be used to document what interval in the data quantitatively reflect the stated properties that at present are declared as a fact. The authors seem to be unaware that recording such agyrotropy in conjunction with other magnetic geometry indicators may already be enough to all reconnection to proceed. There seems to be interest only in the waves for this purpose without considering their role as causal to reconnection or reflexive of the gyro-mechanics that caused the agyrotropy to form in the beginning. This situation is complicated by the heavy aliasing of the plasma measurements by the time variations of the waves being reported. This heavy aliasing also clouds the instability analysis reported to explain the EBW.

Reply:

1. We agree that the pitch-angle distributions can't be used to show agyrotropy. In the previous submission, we used 2D slices of electron distribution functions to present the crescent-shaped agyrotropic electrons. Figure 1 of the new submission (Figure R4) adds one panel to show the agyrotropy measures to show the agyrotropic electrons during the magnetopause crossing. Figure R4i shows the agyrotropy measures \sqrt{Q} defined by *Swisdak et al.* [2016] and $A\phi_e/2$ defined by *Scudder and Daughton* [2008]. In the yellow

region, $A\phi_e/2$ increases significantly in the yellow region, while \sqrt{Q} increase slightly. This is due to the primary difference between these two measures that $A\phi_e/2$ only considers agyrotropy in the plane perpendicular to \mathbf{B} , while \sqrt{Q} measures the full agyrotropy using all components of the electron pressure tensor \mathbf{P}_e . The magnitudes of the measures rely on the relative proportions between the core and the agyrotropic populations [Norgren et al., 2016]. The fact that the measures peak when MMS locate on the magnetosheath side of the neutral line ($B_L < 0$), strongly suggests an electron diffusion region crossing [e.g., Chen et al., 2016; Shay et al., 2016]. We revised Figure 1 of the re-submission and the text to clarify the argument of electron diffusion region.

Figure R4: Magnetopause crossing observed by MMS1. (a) \mathbf{B} . (b) Number density N . (c) \mathbf{V}_i . (d) Ion differential energy flux. (e) \mathbf{V}_e . (f) Electron T_{\parallel} and T_{\perp} . (g) Electron differential energy flux. (h) Electron pitch-angle distribution between 20 eV and 1 keV. (i) Agyrotropy measures \sqrt{Q} and $A\phi_e/2$. (j) \mathbf{E} with frequencies $f < 50$ Hz. The vectors are all presented in LMN coordinate system. The red and blue vertical lines represent the neutral line and the magnetospheric separatrix, respectively. The yellow-shaded region denotes an electron diffusion region crossing.

2. The plasma distribution functions have three dimensions ($[W, \varphi, \theta]$: W for energy, φ for azimuthal angle, and θ for polar angle) in the velocity space. Two-dimensional (2D) slices are widely adopted to present the distribution functions in a particular plane, e.g., $\mathbf{V}_E - \mathbf{V}_{ExB}$ plane. Figure R5 shows an example of how the 2D slices of 3D electron distribution

function are suppressed. The typical frequency of the reported EBWs is 5.7 kHz, and the sampling frequency of the electron distribution functions (7.5 ms) is 133 Hz, which is much lower than the EBWs frequency. Generally, Figure R5b (Fig. 5e of the re-submission) shows the observed background electron distribution function. We add on sub-section with Figure R5 in the *Method* section to answer several questions on the electron distribution functions.

Figure R5: 2D slice of 3D electron distribution function. (a) Skymap of the electron distribution functions with electron velocity of 8.6×10^3 km/s (corresponding to 212 eV). The green, magenta, and red arrowed lines denote the directions of V_E , V_B , and V_{ExB} , respectively. The black circle represents the V_E - V_{ExB} plane perpendicular to the magnetic field \mathbf{B} , and the two dashed circles show the ranges of $\pm 22.5^\circ$ away from the V_E - V_{ExB} plane. The blue arrowed line shows the \mathbf{E}_{max} direction of the EBWs, and the white dot highlights the location with the largest phase-space density of the electron crescent. The intense phase-space densities close to the V_B direction corresponds to the parallel magnetosheath electrons moving towards the X line. The 2D slice of the distribution function in the V_E - V_{ExB} plane (b) is from the averaging of the phase-space densities within $\pm 22.5^\circ$ from the perpendicular plane. The black circle in (b) denotes the average phase-space densities from (a). The black and blue lines show the projected directions with the peak phase-space density of the crescent and \mathbf{E}_{max} .

2. The preparation for the dispersion analysis also appears sketchy to this reviewer. Shouldn't the sufficiency of the eVDF for this modeling be judged based on reproducing the entire measured distribution function before doing an initial value instability analysis? The present text appeals to having matched the perpendicular cut of the distribution as a sufficient test for the initial value simulation. Shouldn't the modeled input crescent distribution for instability analysis exhibit its faithfulness by plotting f_{disp} vs f_{obs} as a Y vs X scatter plot on log, log paper using every velocity pixel measured to demonstrate the realism of the input for the analysis for the dispersion solver. What is the dispersion solver used? It should be referenced.

Reply: The observed electron Bernstein waves are driven by agyrotropic crescent-shaped electrons. We could not find a dispersion solver tool that can include the agyrotropic distribution functions. In this study, we use a *gyrotropic* ring-type distribution to model the observed

crescents, while the rest part is modelled by a combination of a parallel-moving Maxwellian and Maxwellian cores (Fig. R6a). All the parameters are adjusted to have a best fitting of the crescent part and a relatively good fitting of the rest part (e.g., parallel electron beam). The fully kinetic linear dispersion solver developed by K. Min and K. Liu [*Min and Liu, 2015 & 2016*], was used to reveal the generation of ion Bernstein waves by various types of ion distribution functions, e.g., ring, shell and partial shell distribution functions (details can be found in *Min and Liu [2016]*). In this study, K. Min and K. Liu carried out the calculation of dispersion solver, which successfully shows the unstable electron Bernstein modes. The dispersion solver is referenced in the re-submission.

As we discussed in the final part of the draft, the electron Bernstein waves are driven by the agyrotropic electrons in the plane perpendicular to \mathbf{B} via wave-mode coupling between the beam-type model and the fundamental wave modes. That is verified by the fact that another dispersion relation calculation without the parallel beam gives similar results of the unstable electron Bernstein mode.

Figure R6: Model electron distribution function for the wave analysis. Electron model distribution on (a) the $V_{||} - V_{\perp}$ plane and (b) the perpendicular plane. The agyrotropic electron crescent is modelled by a gyrotropic ring-type distribution. Comparison of the observed and model distribution functions along (c) the parallel and (d) the perpendicular directions.

3. Other places: undescribed pitch angles pictures contain overplotted 3 orthogonal L,M,N directions superposed on the 2 independent variables to draw conclusions about who is pointed in which directions.

Reply: Figure R7 (Fig. 3e of the re-submission) shows an example of an electron distribution slice from the average distribution function within $\pm 22.5^\circ$ of the $\mathbf{V}_E - \mathbf{V}_{ExB}$ plane. The \mathbf{V}_E and \mathbf{V}_{ExB} directions are not constant during the investigated time interval. To relate the electron distributions with reconnection, the LMN coordinates are projected on the electron distribution

function slices [e.g., Norgren et al., 2016; Wang et al., 2019], or the electron distribution function slices are presented in the constant LMN coordinates [e.g., Chen et al., 2016]. For Figure R7, the **L** direction is 17° away from the exact $\mathbf{V}_E - \mathbf{V}_{ExB}$ plane, and the projected direction **L** in the figure is normalized by 10^4 km/s. In the new submission, we overplot **L** direction on Fig. 3(e) and 3(g) and **M** direction on Fig. 3(f) and 3(h).

Figure R7: 30 ms Electron distribution function slice on $\mathbf{V}_E - \mathbf{V}_{ExB}$ plane with **L** direction projection. Same with Figure 3e of the re-submitted draft.

4. Flat topped electron distributions in the magnetosheath are common; the paper interprets the flattening of the distribution as evidence for the role of EBW in regulating the reconnection process. Because the initial value distribution function is not shown, there is hardly a before and after demonstration that the diffusion process was important.

Reply: The flat-topped electron distributions are common. Figure R8a and the red curve of Figure R8c show an example of the magnetosheath electrons in this analyzed event. The electron distribution functions are nearly isotropic, with an average temperature of 47 eV. The agyrotropic electrons that driving the EBWs and the diffused electrons are inside the reconnected region. Their average temperature is about 200 eV. The large-amplitude EBWs release the free energy of the agyrotropic electrons, and further thermalize the outflow electrons. A numerical simulation is needed to reveal the detailed process of how the EBWs reform the crescents.

Figure R8. Comparison of (a) the magnetosheath electron distribution function and (b) one diffused by the EBWs. (c) The red curve is the average distribution function of

magnetosheath electron, and the black curve denotes the 1D diffused distribution function along the EBWs direction.

5. Isn't the 7.5 ms FPI electron observations heavily aliased in the EBW interval of 16 ms? If so how is there adequate information for showing that it is unstable using initial value instability measures? Clearly the waves are large amplitude, but the argument about their being important for formation of the diffusion region or field line erosion escapes this reviewer. The instability analysis attempted presupposes the agyrotropic distribution is created first and the EBW evolves from it. Isn't the formation of the agyrotropic distribution of this type already capable of disrupting frozen flux conservation? To advance the argument it would appear necessary to comment on the relative importance of $\text{Div } P_e$ effects seen and those of these waves. From the abstract one gets the impression that the paper had assayed this competition; disappointingly it has not.

Reply: The 7.5 ms FPI electron data product, verified by *Rager et al.* [2018], can show much more detailed energy conversion of the dayside magnetopause reconnection [e.g., *Burch et al.*, 2018]. In this study, the 7.5 ms FPI electrons don't change substantially in the EBW interval of 16 ms and the intervals nearby, which suggests that the 7.5 ms electron distribution functions are not aliased during the EBW observations. The frequencies of the EBWs are from 3.8 kHz to 10.5 kHz, and the sampling frequencies of the 30 ms (e.g, Fig. R7) and 7.5 ms (e.g, Fig. R5b) electron distribution functions are 33 Hz and 133 Hz. Fast Plasma Investigation (FPI) observes the background electron distribution functions around the EBWs interval. 7.5 ms can resolve the electron distribution functions just before and during the large-amplitude EBWs. The agyrotropic electron distribution before the EBWs has sufficient free energy to drive the observed waves.

2D reconnection with agyrotropic distributions does disrupt the magnetic flux frozen conservation. The agyrotropic electron distribution functions, that drive the EBWs, are electron outflow from the X line. Those unstable distribution functions generate waves to release the free kinetic energy. MMS observed the unstable and diffused electron distributions due to EBWs, which suggests that the large-amplitude EBWs can change the electron pressure and modify the balance of reconnection electric field. In this event, the observed nature of magnetic reconnection is already the result of mixture of all the possible effects. It is difficult to reveal the diffusion details of the EBWs effect separately from the data. A reconnection simulation using a particle-in-cell (PIC) model may help us to quantify the EBWs effects and $\nabla \cdot P_e$ effects in and near electron diffusion regions, and it is very challenging to carry out such a simulation. Besides, a kinetic simulation using the initial value conditions [e.g, *Muschiatti and Lembege*, 2013] can show the linear and nonlinear processes of the EBWs generations and quantify the diffusion process due to the EBWs effect. K. Liu (one of the co-authors) is considering the initial value way using a 2D PIC model.

6. p 9 typescript: "The crescent shaped electron distribution [not shown] is close to the $\mathbf{E} \times \mathbf{B}$ direction" This is difficult to swallow since the distribution lives in 3-dimension velocity space and is sub sonic, so in what sense is this 4 surface close to a direction?

Reply: The crescent shaped electron distribution is presented by Figure 4(e) of the previous submission and by Figure 5(e) of this resubmission. We used the 7.5 ms averaged (background) \mathbf{E} and \mathbf{B} fields to determine the \mathbf{V}_E and $\mathbf{V}_{E \times B}$ directions. As shown in Fig. R5, the major part of crescent locates near the $\mathbf{V}_{E \times B}$ direction (red arrowed line) in three-dimension velocity space. One

subsection with Figure R5 is added to clarify how we present and interpret the electron distribution functions.

7. Recommendation: The 25 authors can surely marshal the experimental argument more cogently than in this submitted manuscript. To the principal author: give the submitted manuscript to 3 subgroups of authors for comment about its clarity. Take their comments seriously and revise to alleviate issues like those above.

Reply: Before the first submission and this re-submission, the manuscripts were sent to all the co-authors, and all the received comments and suggestions were seriously considered.

8. This is an interesting topic, but that does not mean its is publishable in its present form. The exposition of the paper's experimental data seems less conversant with clear description of the necessary plasma supporting information. The graphics for, and diagnosis of the crescent distribution is absent or presently unintelligible. Seems unlikely that authors Rager-Giles on the authorship list have approved the submitted article. The article's claim for importance in the electron diffusion region has not separated its role as progenitor or reactor to the agyrotropy, contrary to the impression of the last line in the abstract.

Reply: In the resubmission, we revise all the figures, the last sentence of the abstract, and lots part of the text. One subsection is added in the *Method* section to explain the slices of the electron distribution functions. During the analysis and preparation of this manuscript, the FPI team, including Dr. Rager, Dr. Giles, and Dr. Gershman, provides lots of support to help us understand and visualize the electron data. The FPI co-authors approved the previous and current submissions.

The electrons are energized in the electron diffusion region, and the outflow electron distributions near EDR have agyrotropic features at the electron-scale boundaries. The EBWs driven by the agyrotropic electrons release the free energy and thermalize electrons. This could change the electron pressure and modify the balance of reconnection electric field. We revised the draft to clarify this point.

References:

1. Burch, J. L. et al. (2018). Localized oscillatory energy conversion in magnetopause reconnection. *Geophysical Research Letters*, 45, 1237–1245.
2. Chen, L.-J. et al. (2016) Electron energization and mixing observed by mms in the vicinity of an electron diffusion region during magnetopause reconnection. *Geophys. Res. Lett.* 43, 6036– 6043.
3. Min, K. & Liu, K. (2015) Fast magnetosonic waves driven by shell velocity distributions. *J. Geophys. Res. Space Physics* 120, 2739–2753.
4. Min, K. & Liu, K. (2016) Understanding the growth rate patterns of ion Bernstein instabilities driven by ring-like proton velocity distributions. *J. Geophys. Res. Space Physics* 121, 3036–3049.
5. Muschietti, L. & Lembège B. (2013), Microturbulence in the electron cyclotron frequency range at perpendicular supercritical shocks. *J. Geophys. Res. Space Physics* 118, 2267–2285.

6. Norgren, C. et al. (2016) Finite gyroradius effects in the electron outflow of asymmetric magnetic reconnection. *Geophys. Res. Lett.* 43, 6724–6733.
7. Rager, A. C. et al. (2018) Electron crescent distributions as a manifestation of diamagnetic drift in an electron scale current sheet: Magnetospheric multiscale observations using new 7.5 ms fast plasma investigation moments. *Geophys. Res. Lett.* 45, 578–584.
8. Scudder, J. & Daughton (2008), W. illuminating electron diffusion regions of collisionless magnetic reconnection using electron agyrotropy. *J. Geophys. Res. Space Physics* 113, A06222.
9. Shay, M. A. et al. (2016), Kinetic signatures of the region surrounding the X line in asymmetric (magnetopause) reconnection, *Geophys. Res. Lett.*, 43, 4145–4154.
10. Swisdak, M. (2016) Quantifying gyrotropy in magnetic reconnection. *Geophys. Res. Lett.* 43, 43–49.
11. Wang S. (2019) Ion behaviors in the reconnection diffusion region of a corrugated magnetotail current sheet. *Geophys. Res. Lett.*, just accepted.

Reviewers' comments:

Reviewer #1 (Remarks to the Author):

The authors should acknowledge in the paper, near line 182, that there is a discrepancy between their simulations and observations regarding where the wave powers peak between the harmonics, if they like including their speculation about the reason for it. A sentence is sufficient, but it shouldn't go unacknowledged, as it is too obvious a feature.

Aside from this minor comment, paper is much improved as to clarity of presentation and is suitable and appropriate for publication.

Reviewer #2 (Remarks to the Author):

The reply addressed all my comments, thus I support to publish this paper.

Reviewer #1 additional comments during consultation (Remarks to the Author):

I think one of the main problems with this paper is that it's way too terse. I assume that the authors are working against a fairly severe page/figure number limitation? I think clarity and reproducibility of the study suffer because of the extreme terseness.

On my comments, I was a little disappointed to see in the rebuttal, for example, that they claim in two places that "filtering information...is added in the captions and the figures," when, as far as I can tell, the information is added in the figures only. This is annoying because it suggests the rebuttal can't really be trusted, every little thing has to be checked.

The other two things on my comments: it's nice to see Figure R2 and accompanying discussion in the rebuttal, but at the very least the criterion $E_{\perp} > E_{\max}/e^2$ should be in the text, not just in the communication to the referee. The idea here is, to try to document things so that someone has a chance of reproducing the result.

Finally, in the rebuttal the authors admit my point that the data, in contrast to their theory, show peaks between rather than at gyroharmonics, and they suggest a reason for the discrepancy and a future study to clarify it, but none of this makes it into the paper. I think it should, because my observation is rather obvious and will occur to other readers.

On to referee 3---Referee 3 is a lot more critical of the paper than I am.

His/her principal issue is whether the authors overstepped in implying that the EBW's are an agent of causing the reconnection versus the agyrotropicity (which is purported to also cause the EBWs) being the main cause. The referee wants this subject explored more, and the authors basically argue that it requires difficult simulations beyond the scope of the paper. None of their argumentation makes it into the paper. I don't have access to the previous version to see what the original statement was in the abstract, but it does appear the authors may have toned down their conclusions, perhaps sufficiently to satisfy referee 3. They say that the EBW contribute to the cross field diffusion but don't suggest they are the predominant factor. I suspect referee 3 would like to see some of the discussion in the rebuttal, concerning the competing effects and the need for detailed PIC simulation to sort it out, make it into the paper rather than just in the rebuttal to the referee. Adding some of this discussion would address referee 3s biggest concern that the paper kind of implicitly suggests a larger role for the waves than it really proves.

Another issue raised by referee 3 is need for a more details of how the initial conditions for the instability calculation were determined.

Again, the authors provide a lot of information in the rebuttal (Fig R6 and second paragraph above it), none of which makes it into the paper. I would concur with referee 3 in wanting to see some of this in the paper---again, so that readers have a fighting chance of reproducing the work.

Referee 3 also brings up some technical matters about aliasing of electron distribution measurements and terminologies and determinations of directions related to 3D distributions. I guess I feel the authors have probably sufficiently addressed these points in the rebuttal. Additional wording in the paper could help clarify these things.

Upshot is: I feel the authors should include somewhat more in the paper.

Does this journal allow submission of auxiliary materials? That could be a solution if strict page or line number limits prevent lengthening of the paper. If the authors are not up against such strict page or line number limits, they really should expand a little bit to make the paper clearer.

We appreciate all the comments and suggestions, and have addressed all the points and revised the manuscript. The detailed answers are listed below.

Reviewer #1 (Remarks to the Author):

1. The authors should acknowledge in the paper, near line 182, that there is a discrepancy between their simulations and observations regarding where the wave powers peak between the harmonics, if they like including their speculation about the reason for it. A sentence is sufficient, but it shouldn't go unacknowledged, as it is too obvious a feature.

Reply: Yes. It is obvious that there is a clear difference between the MMS observations and the linear theory about the wave frequencies regarding to the gyro-harmonics. In the re-submission, we point out this difference with our speculation.

2. Aside from this minor comment, paper is much improved as to clarity of presentation and is suitable and appropriate for publication.

Reply: Thank you.

Reviewer #2 (Remarks to the Author):

The reply addressed all my comments, thus I support to publish this paper.

Reply: Thank you.

Reviewer #1 additional comments during consultation (Remarks to the Author):

1. I think one of the main problems with this paper is that it's way too terse. I assume that the authors are working against a fairly severe page/figure number limitation? I think clarity and reproducibility of the study suffer because of the extreme terseness.

Reply: In this re-submission, we add more sentences to address all the points raised by the referee. The difference between the MMS data and linear theory analysis is added, and the discussion part is revised to tone down part of our conclusion. We add one sub-section in *Method* section with an auxiliary material to show the details of the electron model distribution function for the linear wave analysis, so that the readers can reproduce our study. The captions of Figure 3, 4 and 5 are modified to include the filtering information and the criteria for selecting the EBWs interval. The quality of the 7.5 ms data is clarified in the draft.

2. On my comments, I was a little disappointed to see in the rebuttal, for example, that they claim in two places that "filtering information...is added in the captions and the figures," when, as far as I can tell, the information is added in the figures only. This is annoying because it suggests the rebuttal can't really be trusted, every little thing has to be checked.

Reply: In the previous re-submission, we added all the filtering information in all the figures using E field and the caption of Figure 1, which can give the idea that 50 Hz is the frequency to divide the low-frequency and high-frequency parts of the electric field **E**. In this re-submission, all the captions of figures with **E** data include the filtering information to avoid any potential misunderstanding. Also, all the details are checked carefully.

3. The other two things on my comments: it's nice to see Figure R2 and accompanying discussion in the rebuttal, but at the very least the criterion $E_{\perp} > E_{\max}/e^2$ should be in the text, not just in the communication to the referee. The idea here is, to try to document things so that someone has a chance of reproducing the result. Finally, in the rebuttal the authors admit my point that the data, in contrast to their theory, show peaks between rather than at gyroharmonics, and they suggest a reason for the discrepancy and a future study to clarify it, but none of this makes it into the paper. I think it should, because my observation is rather obvious and will occur to other readers.

Reply: How we selected the EBWs interval is added in the caption of Figure 4. It is true that there is a clear difference between the MMS observations and the linear theory about the wave frequencies regarding to the gyro-harmonics. In this revised manuscript, we point out this difference with our speculation.

4. On to referee 3---Referee 3 is a lot more critical of the paper than I am. His/her principal issue is whether the authors overstepped in implying that the EBW's are an agent of causing the reconnection versus the agyrotropicity (which is purported to also cause the EBWs) being the main cause. The referee wants this subject explored more, and the authors basically argue that it requires difficult simulations beyond the scope of the paper. None of their argumentation makes it into the paper. I don't have access to the previous version to see what the original statement was in the abstract, but it does appear the authors may have toned down their conclusions, perhaps sufficiently to satisfy referee 3. They say that the EBW contribute to the cross field diffusion but don't suggest they are the predominant factor. I suspect referee 3 would like to see some of the discussion in the rebuttal, concerning the competing effects and the need for detailed PIC simulation to sort it out, make it into the paper rather than just in the rebuttal to the referee. Adding some of this discussion would address referee 3s biggest concern that the paper kind of implicitly suggests a larger role for the waves than it really proves.

Reply: The discussion section of the manuscript is revised to tone down our conclusion of this study. The particle-in-cell simulation is needed to quantify the diffusion effect of the EBWs. Dokgo et al. [2019] reported a detailed theoretical and numerical analysis of the upper-hybrid waves observed in

an electron diffusion region of magnetotail reconnection. They have interest to use their PIC model to analyze the linear-nonlinear process and quantify the effect of EBWs.

5. Another issue raised by referee 3 is need for a more details of how the initial conditions for the instability calculation were determined. Again, the authors provide a lot of information in the rebuttal (Fig R6 and second paragraph above it), none of which makes it into the paper. I would concur with referee 3 in wanting to see some of this in the paper---again, so that readers have a fighting chance of reproducing the work.

Reply: We add one sub-section with an auxiliary material in *Method* section to present the details of the electron model distribution function for the dispersion relation analysis, so that the readers can reproduce our study.

6. Referee 3 also brings up some technical matters about aliasing of electron distribution measurements and terminologies and determinations of directions related to 3D distributions. I guess I feel the authors have probably sufficiently addressed these points in the rebuttal. Additional wording in the paper could help clarify these things.

Reply: We add more information in the re-submission to clarify the 7.5 ms electron distribution function data. Also, one sub-section in *Method* section was added to clarify some technical treatment of the three-dimensional distribution functions.

7. Upshot is: I feel the authors should include somewhat more in the paper. Does this journal allow submission of auxiliary materials? That could be a solution if strict page or line number limits prevent lengthening of the paper. If the authors are not up against such strict page or line number limits, they really should expand a little bit to make the paper clearer.

Reply: Thank you so much for the suggestions here and during the past reviewing process. As been replied in the beginning, we add more information (including an auxiliary material) in several parts of the draft to address all the unclear points mentioned by the reviewer.

Reference:

Dokgo, K., Hwang, K.-J., Burch, J. L., Choi, E., Yoon, P. H., Sibeck, D. G., & Graham, D. B. (2019). High-frequency wave generation in magnetotail reconnection: Nonlinear harmonics of upper hybrid waves. *Geophysical Research Letters*, 46. Doi: 10.1029/2019GL083361.

Reviewers' comments:

Reviewer #1 (Remarks to the Author):

I re-read all the comments and rebuttals and then re-read the paper carefully from beginning to end, including figure captions.

I find the paper much improved, and I think the concerns of referee 3 have been addressed.

I have some remaining comments on presentation, which I think the authors should find straightforward to address:

lines 84-85: many readers will not be familiar with these anisotropy measures, and while they can go to the reference for detailed descriptions, it would seem fair to give them a rough idea, in a sentence, what these parameters are and, especially, whether the observed values (up to 0.05) are high or low and relative to what (i.e., provide some context for the reader for these numbers)

Here and at line 96---from the time series of these parameters one might conclude that the agyrotropivities are sporadic. The parameters bounce up and down. Is this the correct interpretation? Please clarify either way; if it is sporadic, say so, and if not, say why it is not, despite the variations in these parameters. (Maybe this is partly why referee 3 focussed in on the possibility of aliasing in the distributions?)

line 93, the reference seems inadequately described for me to track down. Also, in case it is an entire book (I recall an entire ISSI or ESA volume on this topic), it would be too broad to be useful. Can the authors provide a better reference? I assume there is nothing unduly difficult or sophisticated about this analysis; otherwise a bit more information in the paper would be appropriate.

line 114: state whether this is the combined E-field of all the wave modes, or filtered to analyze a single mode.

line 124, on outer magnetosphere EBW's, only one MMS paper is cited, but I thought there was a wealth of pre-MMS papers on EBWs in the outer magnetosphere. I'm thinking of numerous papers detailing " $n+1/2$ waves" and their possible causes. I think EBW was among the causes speculated for

these waves, going back to early days. Some contact with that literature here might be appropriate, rather than just citing one MMS paper.

lines 140-144, I suggest a slight rewording something like, "If one assumes the direction of phase velocity is..., the the frequency of peak power...corresponds to a wavelength of 1.4 km, which is comparable to..."

line 149: it's unclear what "that" refers to (that="the measured distribution"?)

lines 170-172: it's great that the authors have included these ideas, but the presentation is somewhat confusing saying one idea is probably the case but then the other completely different idea could be responsible. It might be clearer to say "Two possibilities to explain this are...idea 1 and idea 2." If one of the ideas is much more likely, keep something more like the current wording but indicate why.

lines 203-204: I think it would be more precise to say something like: MMS observed distributions just before the wave event which are presumed to be close to the unstable ones, and during the event which are presumed to be the diffused ones. In other words, say what the actual measurement was, as well as the interpretation.

line 212: Provide more details here. Does this number come from Figure 6 of the reference, assuming a wave power of a few times $10^{-5} \text{ V}^2/\text{m}^2$?

which is related to one final clarification needed:

Figure 4f: There's a problem with the label of Figure 4f, in that the reader cannot tell whether the scale is linear or log, and either way, what is the range of the scale on the vertical axis. Please give another value besides 10^0 so the readers know how to read values off of the plot.

We appreciate all the comments and suggestions from the reviewer, and addressed all the points in the revised manuscript. The detailed answers are listed below.

Reviewer #1 (Remarks to the Author):

1. lines 84-85: many readers will not be familiar with these anisotropy measures, and while they can go to the reference for detailed descriptions, it would seem fair to give them a rough idea, in a sentence, what these parameters are and, especially, whether the observed values (up to 0.05) are high or low and relative to what (i.e., provide some context for the reader for these numbers)

Reply: We add some descriptions to explain the basic idea of the agyrotropy measure. Typical values are about 0.1 around the electron diffusion region at the magnetopause [Norgren *et al.*, 2016; Graham *et al.*, 2017]. Also, one agyrotropy measure is enough for presenting the agyrotropy distribution in this event, so we only show one agyrotropy measure \sqrt{Q} in the revision.

2. Here and at line 96---from the time series of these parameters one might conclude that the agyrotropies are sporadic. The parameters bounce up and down. Is this the correct interpretation? Please clarify either way; if it is sporadic, say so, and if not, say why it is not, despite the variations in these parameters. (Maybe this is partly why referee 3 focused in on the possibility of aliasing in the distributions?)

Reply: A background of the agyrotropy measure, 0.0124, is estimated from the MMS data before the EDR crossing. We can find clear agyrotropy \sqrt{Q} enhancement during the EDR crossing, especially before the EBWs interval. We revised the draft.

3. line 93, the reference seems inadequately described for me to track down. Also, in case it is an entire book (I recall an entire ISSI or ESA volume on this topic), it would be too broad to be useful. Can the authors provide a better reference? I assume there is nothing unduly difficult or sophisticated about this analysis; otherwise a bit more information in the paper would be appropriate.

Reply: We cite the chapter of the ISSI book for the reference of the timing method, and change the reference for the minimum variance analysis (MVA) method.

4. line 114: state whether this is the combined E-field of all the wave modes, or filtered to analyze a single mode.

Reply: The wave E-field is filtered with frequencies $f > 50$ Hz, meaning all Bernstein modes are included. The draft is revised to fix the ambiguity.

5. line 124, on outer magnetosphere EBW's, only one MMS paper is cited, but I thought there was a wealth of pre-MMS papers on EBWs in the outer magnetosphere. I'm thinking of numerous papers detailing "n+1/2 waves" and their possible causes. I think EBW was among the causes speculated for these waves, going back to early days. Some contact with that literature here might be appropriate, rather than just citing one MMS paper.

Reply: We change the references to cite papers on EBWs using pre-MMS spacecraft data.

6. lines 140-144, I suggest a slight rewording something like, "If one assumes the direction of phase

velocity is..., the frequency of peak power...corresponds to a wavelength of 1.4 km, which is comparable to..."

Reply: We revised the paragraph to make this more logical.

7. line 149: it's unclear what "that" refers to (that="the measured distribution"?)

Reply: That refers to the measured electron distribution. The text is revised to make it clearer.

8. lines 170-172: it's great that the authors have included these ideas, but the presentation is somewhat confusing saying one idea is probably the case but then the other completely different idea could be responsible. It might be clearer to say "Two possibilities to explain this are...idea 1 and idea 2." If one of the ideas is much more likely, keep something more like the current wording but indicate why.

Reply: The sentence is revised. We want to keep these two possibilities and more work is needed to confirm.

9. lines 203-204: I think it would be more precise to say something like: MMS observed distributions just before the wave event which are presumed to be close to the unstable ones, and during the event which are presumed to be the diffused ones. In other words, say what the actual measurement was, as well as the interpretation.

Reply: The text is revised to make the sentence clearer.

10. line 212: Provide more details here. Does this number come from Figure 6 of the reference, assuming a wave power of a few times $10^{-5} \text{ V}^2/\text{m}^2$?

Reply: The diffusion coefficient in the draft comes from Eq. (4) and (9) of the reference, assuming a EBWs amplitude of 60 mV/m, while Figure 6 of the reference [LaBelle and Treumann, 1988] used the plasma and magnetopause parameters from LaBelle et al. [1987]. More details are provided in the revision.

11. Figure 4f: There's a problem with the label of Figure 4f, in that the reader cannot tell whether the scale is linear or log, and either way, what is the range of the scale on the vertical axis. Please give another value besides 10^0 so the readers know how to read values off of the plot.

Reply: The Y axis of Figure 4f uses log scale, and we add more ticks besides ' 10^0 ' to make the magnitudes of the wave power spectral density clearer.

References:

1. Graham, D. B. et al. Instability of agyrotropic electron beams near the electron diffusion region. *Phys. Rev. Lett.* 119, 025101 (2017).
2. LaBelle, J., Treumann, R. A., Haerendel, G., Bauer, O. H., Paschmann, G., Baumjohann, W., Lühr, H., Anderson, R. R., Koons, H. C., and Holzworth, R. H. (1987), AMPTE IRM observations of waves associated with flux transfer events in the magnetosphere, *J. Geophys. Res.*, 92(A6), 5827– 5843, doi:10.1029/JA092iA06p05827.
3. Labelle, J. & Treumann, R. A. Plasma waves at the dayside magnetopause. *Space Science Reviews* 47, 175–202 (1988).
4. Norgren, C. et al. Finite gyroradius effects in the electron outflow of asymmetric magnetic reconnection. *Geophys. Res. Lett.* 43, 6724–6733 (2016).

REVIEWERS' COMMENTS:

Reviewer #1 (Remarks to the Author):

The authors have responded to my comments, so from my point of view the paper is acceptable for publication